# Cohesin controls intestinal stem cell identity by maintaining association of Escargot with target promoters

Aliaksandr Khaminets[1]*, Tal Ronnen-Oron[2], Maik Baldauf[1], Elke Meier[1], Heinrich Jasper[1,2,3]*

[1]Leibniz Institute on Aging – Fritz Lipmann Institute (FLI), Jena, Germany; [2]Buck Institute for Research on Aging, Novato, United States; [3]Immunology Discovery, Genentech, Inc, South San Francisco, United States

**Abstract** Intestinal stem cells (ISCs) maintain regenerative capacity of the intestinal epithelium. Their function and activity are regulated by transcriptional changes, yet how such changes are coordinated at the genomic level remains unclear. The Cohesin complex regulates transcription globally by generating topologically-associated DNA domains (TADs) that link promotor regions with distant enhancers. We show here that the Cohesin complex prevents premature differentiation of *Drosophila* ISCs into enterocytes (ECs). Depletion of the Cohesin subunit Rad21 and the loading factor Nipped-B triggers an ISC to EC differentiation program that is independent of Notch signaling, but can be rescued by over-expression of the ISC-specific escargot (esg) transcription factor. Using damID and transcriptomic analysis, we find that Cohesin regulates Esg binding to promoters of differentiation genes, including a group of Notch target genes involved in ISC differentiation. We propose that Cohesin ensures efficient Esg-dependent gene repression to maintain stemness and intestinal homeostasis.

*For correspondence:
akhaminets@gmail.com (AK);
jasper.heinrich@gene.com (HJ)

## Introduction

Intestinal stem cells (ISCs) contribute to epithelial homeostasis in the gastrointestinal tract due to their capacity to proliferate and renew old or damaged tissue (*Casali and Batlle, 2009*; *Biteau et al., 2011*; *Signer and Morrison, 2013*; *Barker, 2014*). ISCs self-renew and supply cells that differentiate into different lineages needed to perform all essential functions of the intestinal epithelium (*Jiang and Edgar, 2011*; *Li and Jasper, 2016*). ISC maintenance and proliferative activity thus influences intestinal homeostasis, as well as the general fitness and healthy life span (or health-span) of the organism (*Li and Jasper, 2016*). While many intrinsic and extrinsic signals that govern ISC maintenance and function have been described (*Casali and Batlle, 2009*; *Biteau et al., 2011*; *Signer and Morrison, 2013*; *Barker, 2014*), little is known about how different elements of these regulatory networks interact to coordinate ISC proliferation and differentiation programs.

*Drosophila melanogaster* has proven to be a powerful system to dissect conserved signaling pathways, cellular networks, genetic, epigenetic and environmental factors influencing ISC function (*Jasper, 2015*; *Li and Jasper, 2016*). The intestinal epithelium in the fly midgut consists of ISCs, enteroblasts (EBs), enterocytes (ECs) and enteroendocrine (EEs) cells (*Miguel-Aliaga et al., 2018*). ISCs are the main mitotic cells in this tissue and can give rise to all cell types performing essential secretory, absorptive and immune roles (*Ayyaz and Jasper, 2013*; *Li and Jasper, 2016*; *Liu and Jin, 2017*). A number of highly conserved and well-studied signaling cascades control ISC proliferation and differentiation in response to outside stimuli (*Li and Jasper, 2016*; *Miguel-Aliaga et al., 2018*). Infection or stress, for example, triggers ISC proliferation through the activation of JAK/STAT, JNK, and EGFR signaling pathways (*Biteau et al., 2008*; *Jiang and Edgar, 2009*; *Jiang et al., 2009*;

*Buchon et al., 2010*; *Lin et al., 2010*; *Liu et al., 2010*; *Jiang et al., 2011*). Notch activity limits ISC proliferation and is required for EB differentiation into ECs (*Ohlstein and Spradling, 2007*; *Kapuria et al., 2012*; *Guo and Ohlstein, 2015*). Intrinsic regulators of ISC maintenance include the master transcription factor Escargot (Esg), which positively regulates expression of stemness genes, but inhibits expression of differentiation factors, including of the transcription factor Nubbin/Pdm1 (*Korzelius et al., 2014*; *Loza-Coll et al., 2014*). It remains unclear how these and other extrinsic and intrinsic factors interact with the chromatin status of ISCs, and how chromatin regulation influences ISC activity and differentiation.

Chromatin is organized in a hierarchical fashion into highly dynamic domains (*Filion et al., 2010*; *Sexton and Cavalli, 2015*). Architectural proteins shape DNA and together with other epigenetic mechanisms control transcriptional responses in a cell type-dependent manner (*Peric-Hupkes et al., 2010*; *Phillips-Cremins et al., 2013*; *Adam et al., 2015*; *Beagan et al., 2016*; *Huang et al., 2016*; *Neems et al., 2016*; *Schmitt et al., 2016*). Cohesin, a ring-like structure formed by a complex of proteins (Smc1, Smc3, Rad21 and SA1/SA2 proteins) that interact with DNA, is a key element in genome organization (*Peters et al., 2008*; *Feeney et al., 2010*; *Singh and Gerton, 2015*). During mitosis, Cohesin encircles paired chromosomes and holds them together until late anaphase, when Rad21 is cleaved by the protease Separase to allow chromosome segregation towards centrosomes (*Haarhuis et al., 2014*; *Morales and Losada, 2018*). This ensures synchronous resolution and correct inheritance of chromatids and protects cells from aneuploidy (*Onn et al., 2008*; *Haarhuis et al., 2013*; *Singh and Gerton, 2015*). While only a minor amount (up to 13%) of Cohesin has been reported to be sufficient for chromosome cohesion (*Heidinger-Pauli et al., 2010*) most of Cohesin appears to be involved in chromatin organization in interphase cells via organization of topologically-associated domains (TADs) by means of loop extrusion (*Dekker and Mirny, 2016*; *Rao et al., 2017*; *Wutz et al., 2017*; *Yuen and Gerton, 2018*). Within TADs, Cohesin interacts with transcription factors and brings promotors into proximity with enhancers (*Schaaf et al., 2013*; *Merkenschlager and Nora, 2016*; *Novo et al., 2018*). Cohesin deficiency results in the elimination of TADs and alters transcription, supporting its crucial function in gene expression (*Pauli et al., 2010*; *Dorsett and Ström, 2012*; *Zuin et al., 2014*; *Lupiáñez et al., 2015*; *Rao et al., 2017*). Cohesin deficiency has further been shown to cause prominent changes in genomic organization, leading to transcription being dependent on local transcriptional elements and cryptic promotors rather than on distant enhancer elements (*Schwarzer et al., 2017*).

In the course of normal differentiation, as well as under stress conditions, stem cells reorganize their chromatin to adapt transcription to differentiation and activation programs, and to achieve new cellular functions (*Cantone and Fisher, 2013*; *Rinaldi and Benitah, 2015*; *Adam and Fuchs, 2016*). Such chromatin remodeling happens in embryonic and neural stem cells during differentiation into other cell types (*Pękowska et al., 2018*), precipitating dramatic transcriptional changes. Whether and how DNA conformation and higher order chromatin structures also regulate stemness and lineage determination in SCs of barrier epithelia, however, has not been explored. In this study, we investigate how Cohesin-dependent chromatin organization regulates ISCs using the fly intestine as a model. We define a transcriptional program mediated by Cohesin and find that Cohesin is essential for ISC maintenance by regulating the transcriptional output of Esg.

## Results

### Rad21 regulates ISC proliferation and differentiation

The Rad21 subunit of the Cohesin complex has been shown to act as a tumor suppressor and promoter of pro-inflammatory myeloid polarization in the hematopoietic system, while acting to preserve stemness and prevent differentiation of progenitor cells in the epidermis (*Mullenders et al., 2015*; *Noutsou et al., 2017*; *Cuartero et al., 2018*). These observations indicate that Cohesin may selectively regulate different types of somatic stem cells, and that further characterization of its role in stem cell behavior in vivo is warranted. Using the *Drosophila* ISC lineage as a model, we asked whether loss of Cohesin would influence stem cells in a barrier epithelium. We performed lineage tracing using the esg-FlpOut system (*Jiang et al., 2009*), which allows temperature-inducible Flp-mediated excision of a transcriptional STOP sequence in ISCs to achieve inherited expression of Gal4, and as a consequence expression of UAS-linked transgenes, in ISCs and their daughter cells.

Using this system, knock down of Rad21 (using two independent shRNA constructs targeting Rad21) resulted in clones mostly containing individual cells with large nuclei and expressing Pdm1, indicating that Rad21 knock down triggered ISC differentiation into enterocytes (ECs) (*Lee et al., 2009*; *Beebe et al., 2010*; *Mathur et al., 2010*; *Korzelius et al., 2014*) (*Figure 1A and B*; *Figure 1—figure supplement 1*). Rad21-deficient cells became larger (*Figure 1—figure supplement 2*) and their nuclei were significantly bigger than nuclei of WT ISCs, supporting the notion of their terminal differentiation into ECs (*Figure 1C*). While Cohesin-depleted clones showed a significant increase in Pdm1 positive cells (*Figure 1*), they were not in fact smaller with respect to the numbers of cells per clone. Their size was about equal compared to wild-type clones, with some minor variability in the exact number of cells seen in different guts (*Figure 1—figure supplement 1*). Elimination of the Rad21 immunohistochemistry signal in ISCs after Rad21 knock down confirmed efficient protein downregulation (*Figure 1—figure supplement 3*).

ISC differentiation is normally associated with abrogation of cell proliferation and reduction of mitotic markers. Consistent with ectopic ISC differentiation, loss of Rad21 inhibited ISC proliferation in response to infection by the mild enteropathogen *Erwinia carotovora carotovora 15* (Ecc15), as assessed by the quantification of phospho-Histone H3+ cells in the gut (*Figure 1D*). This premature differentiation phenotype in Rad21 deficient *Drosophila* ISCs is consistent with loss of epidermal progenitor cells in mice due to aberrant differentiation after Rad21 knock down (*Noutsou et al., 2017*).

Rad21/Cohesin is loaded onto chromosomes by the kollerin complex, which includes Nipped-B, and on chromosome arms is removed during mitotic prophase by Polo kinase, likely via phosphorylation of its SA1/SA2 subunits, while the centromeric pool of Cohesin is removed during the metaphase to anaphase transition by Separase (*Hauf et al., 2005*; *Dorsett, 2009*). We asked whether perturbing Rad21 loading or promoting Cohesin release would be sufficient to phenocopy the effects of Rad21 knockdown. To perturb Cohesin loading or release, we knocked down Nipped-B or over-expressed the constitutively active Polo mutant $polo^{T182D}$ (the *Drosophila* homologue of the mammalian $Plk1^{T210D}$ allele that leads to Cohesin release from chromatin while also delaying mitosis) (*Sumara et al., 2002*; *van de Weerdt et al., 2005*) in ISCs, and analyzed differentiation and proliferation (*Figure 2*). Similar to reducing *rad21* expression, these perturbations induced premature ISC differentiation, indicated by ectopic Pdm1 expression (*Figure 2A–C*) and inhibition of mitotic activity (*Figure 2—figure supplement 1*). Interestingly, depletion of the chromatin-shaping and transcriptional repressor and insulator CTCF (CCCTC-binding factor) (*Sexton and Cavalli, 2015*) did not trigger ISC differentiation (*Figure 2A–C*). These data suggest that loading of Rad21/Cohesin onto chromatin is crucial for ISC maintenance and to prevent premature differentiation of ISCs.

It is important to emphasize that ISCs divide at a very low rate in homeostatic conditions (*Figures 1–4*). The esg-FlpOut system we used allows determining the number of divisions that have occurred in our experiments. As shown in *Figures 1A* and *2A*, in control flies, ISCs have for the most part, not divided (single labeled cells) or divided only once (doublets) in the course of the experiments. Rad21 knockdown, Polo activation, or NippedB knockdown resulted in cell differentiation regardless of whether ISCs had divided (clones with more than one labeled cell), or not (single labeled cell).

We further asked whether Rad21 over-expression would influence ISC behavior, or whether excessive Rad21 would only affect ISCs during mitosis, where its function in chromosome cohesion is important. Over-expression of Rad21-HA in ISCs using esg::Gal4 was sufficient to induce ISC proliferation in homeostatic conditions, suggesting that Rad21 can act in ISCs, at least in part, independently of chromatid cohesion (*Figure 3A and B*; over-expressed Rad21-HA localized to the nucleus, *Figure 3C*). Similarly, Rad21-HA overexpression in ISCs using esg-FlpOut system induced drastic cell overproliferation with formation of Pdm1-positive ECs (*Figure 3—figure supplement 1*). We therefore hypothesized that Rad21 influences ISC function via transcriptional regulation of proliferation and stemness genes in interphase, similar to its role in the mammalian epidermis (*Noutsou et al., 2017*).

## Mitotic failure impacts cohesin and ISC homeostasis

Rad21 and the Cohesin complex are critical for the accurate distribution of chromosomes during mitosis, and their perturbation has thus been implicated in aneuploidy (*Haarhuis et al., 2014*; *Xu et al., 2014*; *Morales and Losada, 2018*). Aneuploidy has recently been reported to cause ISC differentiation (*Gogendeau et al., 2015*) and also to promote dysplasia (*Resende et al., 2018*).

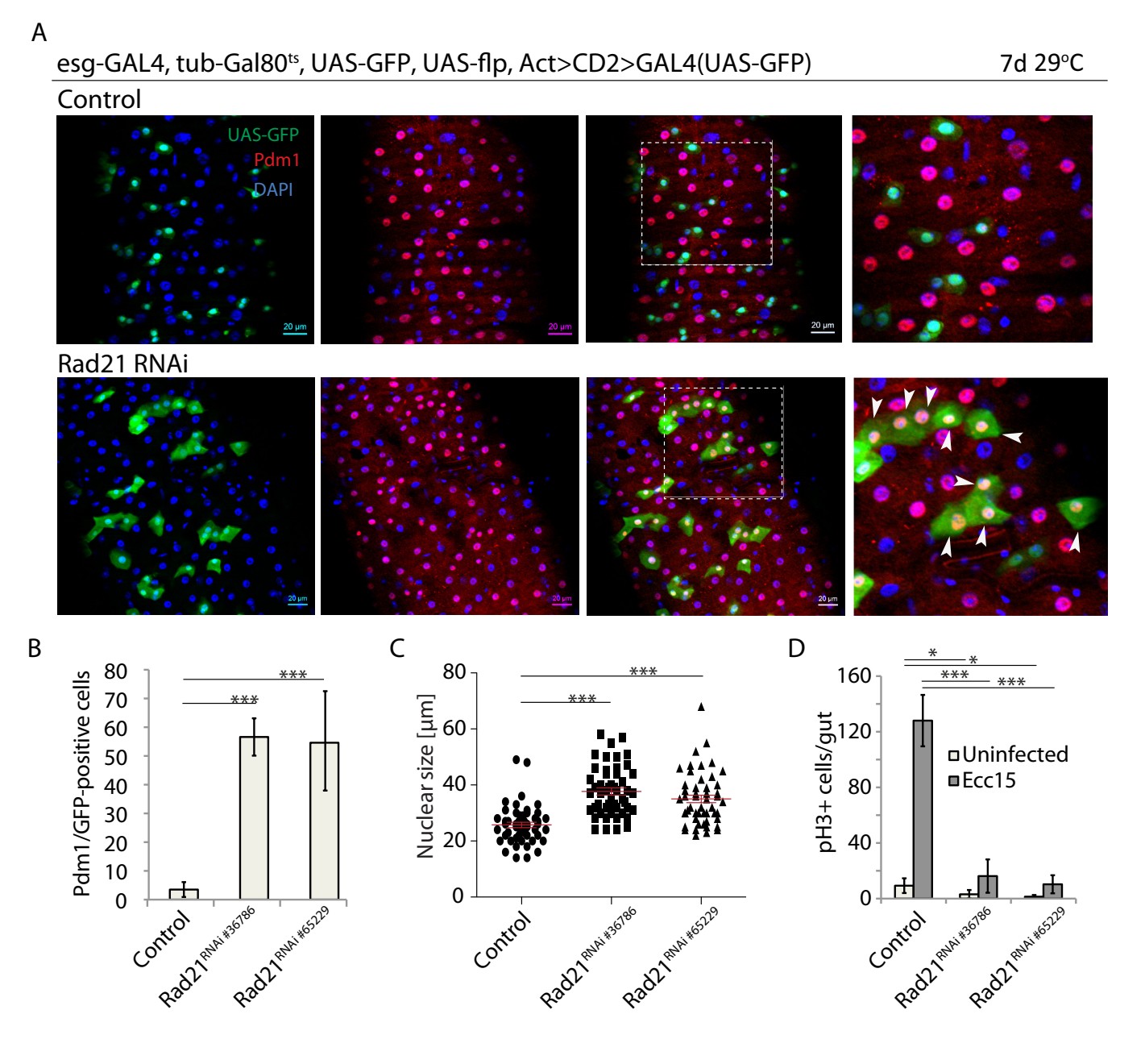

**Figure 1.** Rad21 knock down leads to premature ISC differentiation. (**A**) esg-FlipOut (F/O) midguts expressing UAS-GFP alone (control) or expressing rad21$^{RNAi}$. Samples were stained for GFP and Pdm1. (**B**) Quantification of GFP-positive/Pdm1-positive cells from A. (n = 771, 543 and 389) ANOVA. (**C**) Analysis of nuclei size from A (n = 50). Mann-Whitney Test. (**D**) Quantification of the number of mitotic pH3-positive cells/midgut in the guts expressing UAS-EYFP alone (control) or expressing rad21$^{RNAi}$ with and without Ecc15 infection. (n = 6–8), ANOVA. *p<0.05, ***p<0.001. Differentiated cells are labeled with white arrowheads. Scale bars, 20 μm.

The online version of this article includes the following figure supplement(s) for figure 1:

**Figure supplement 1.** Esg-FlipOut (F/O) midguts expressing UAS-GFP alone (control) or expressing rad21$^{RNAi}$.

**Figure supplement 2.** Esg$^{ts}$ midguts expressing UAS-EYFP alone (control) or expressing rad21$^{RNAi}$.

**Figure supplement 3.** A Esg$^{ts}$ midguts expressing UAS-EYFP alone (control) or expressing rad21$^{RNAi}$.

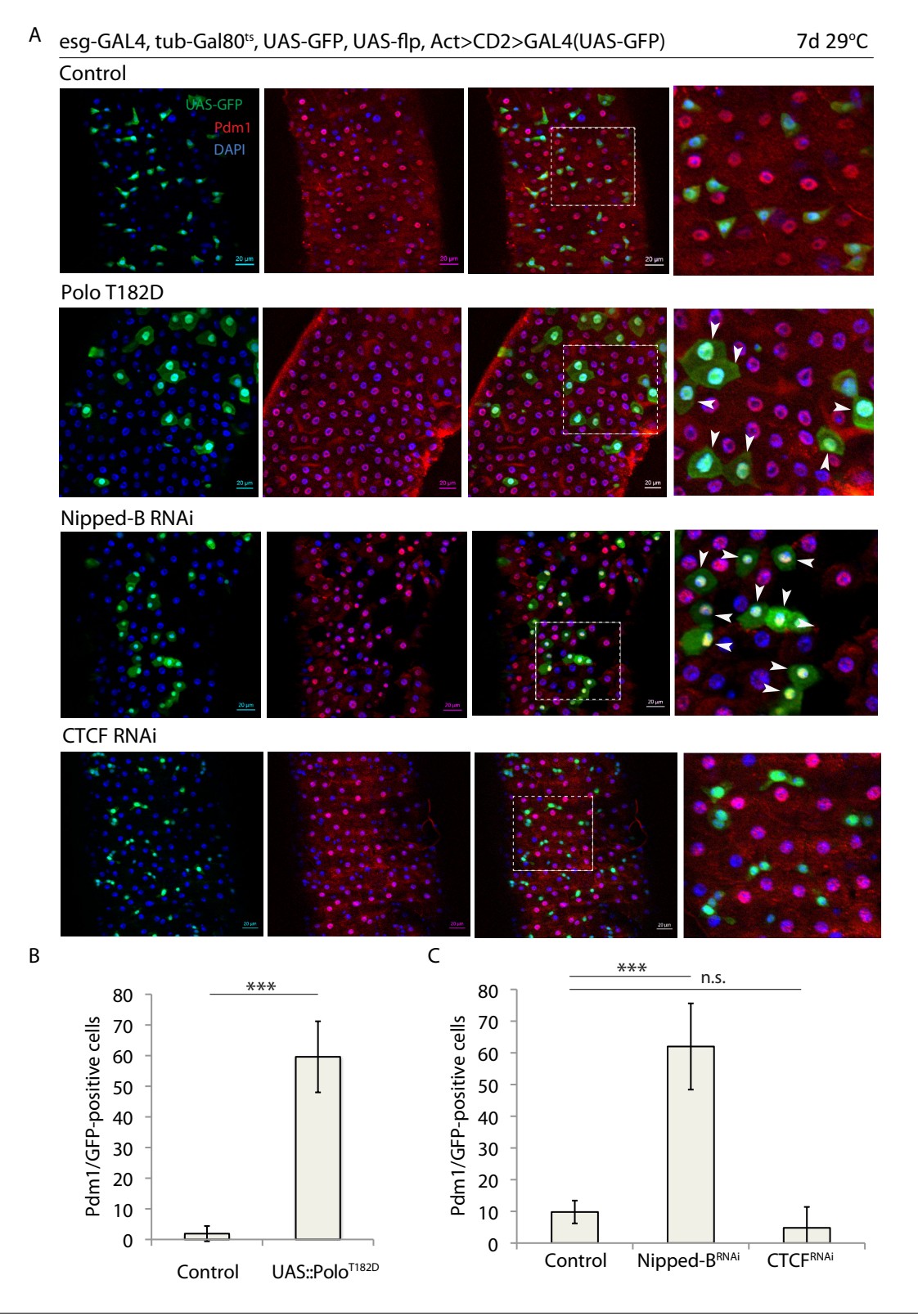

**Figure 2.** Binding of Rad21 to chromatin is crucial for ISC maintenance. (**A**) esg-F/O midguts expressing UAS-GFP alone (control) or expressing UAS-Polo$^{T182D}$, nippedB$^{RNAi}$ or CTCF$^{RNAi}$. Samples were stained for GFP and Pdm1. (**B and C**) Quantification of GFP-positive/Pdm1-positive cells from C. (B n = 640 and 200; C n = 758, 408 and 622), ANOVA. ***p<0.001, n.s. not significant. Differentiated cells are labeled with white arrowheads. Scale bars, 20 μm.

*Figure 2 continued on next page*

*Figure 2 continued*

The online version of this article includes the following figure supplement(s) for figure 2:

**Figure supplement 1.** Overexpression of Polo[T182D] or downregualtion of Nipped-B trigger ISC differentiation.

These previous studies have reported conflicting results regarding the differentiation response of ISCs when aneuploidy is induced by knock down of spindle assembly checkpoint proteins: While depletion of *bub3* was found to induce ISC differentiation and thus loss of ISCs (*Gogendeau et al., 2015*), depletion of BubR1, mad2, or mps1 all resulted in increased ISC proliferation and an accumulation of ISCs/EBs and EEs in the intestinal epithelium (*Resende et al., 2018*). Since depletion of all four factors results in aneuploidy, the differentiation response to Bub3 depletion seems to be a consequence of another function of Bub3, rather than the aneuploidy itself.

To better understand the effects of Rad21 perturbation in ISCs, and to differentiate possible aneuploidy-mediated phenotypes from effects caused by transcriptional changes in interphase, we directly compared the ISC phenotypes caused by aneuploidy with the effects of Rad21 knock down. We knocked down the essential mitosis regulators Cdk1, polo and aurora B (*Godinho and Tavares, 2008*) using esgF/O, and monitored ISC differentiation and proliferation. These perturbations caused premature ISC differentiation, indicated by ectopic Pdm1 and inhibited mitotic activity (*Figure 4A–C*) reminiscent of results in published reports in which the checkpoint kinase Bub3 was knocked down (*Gogendeau et al., 2015*). However, the frequency of Pdm1-positive cells after these perturbations (*Figure 4B*) was prominently lower compared to downregulation of Rad21 (*Figure 1B*). Furthermore, we assessed expression of the ISC marker *Delta* in Rad21 and aurB-depleted ISCs using a Dl-lacZ reporter. AurB RNAi led to a significant reduction in *Delta-lacZ* expression, but Rad21 knock down completely eliminated this expression, indicating a more complete loss of stemness following Rad21 depletion than in other aneuploidy-inducing conditions (*Figure 5A and B*). Since only a minor proportion of ISCs undergo mitosis under homeostatic conditions, these results further point to a cell-cycle independent role of Cohesin in ISCs. Finally, we compared Rad21 localization and expression between ISCs with deregulated mitosis (expressing *polo[T182D]*) and WT cells. Even though Rad21 was localized to the nucleus (*Figure 5—figure supplement 1*) its expression was significantly decreased in *polo[T182D]*–expressing ISCs compared to WT controls (*Figure 5—figure supplement 1*). Thus, cell cycle defects could lead to Rad21 downregulation further triggering ISC differentiation.

## Transcriptional role of Rad21 in regulating ISCs

Rad21 is known to regulate transcription by shaping 3D structure of chromatin and bringing promotors and enhancers in close proximity to each other (*Rudra and Skibbens, 2013*). This is crucial for regulating transcription in interphase and for maintaining transcriptional memory after mitosis (*Yan et al., 2013*). These functions are dependent on recruitment of specific transcription factors and execution of particular transcriptional programs (*Schaaf et al., 2013*; *Merkenschlager and Nora, 2016*; *Novo et al., 2018*). We therefore hypothesized that Rad21 would regulate ISCs via recruitment of specific transcription factors to chromatin and promoting gene expression programs that maintain stemness and inhibit differentiation. To investigate how Rad21 regulates gene expression at a global level in ISCs, we conducted RNAseq analysis of FACS-sorted ISCs in which Rad21 was either downregulated (Rad21 RNAi) or overexpressed (UAS-Rad21-HA) for 3 and 7 days (*Figure 6A–G*). We used a previously described protocol for RNAseq analysis of FACS purified ISCs (*Korzelius et al., 2014*). Our data confirmed Rad21 knock down and overexpression, respectively, in ISCs (*Figure 6E*). We further observed that *rad21* knockdown resulted in significant changes ($q < 0.05$ and $Log_2$ equals or more then 1) in 985 genes against 489 genes in UAS-Rad21-HA samples ($q < 0.05$) with 160-gene overlap between conditions (*Figure 6A and B*, *Figure 6—source data 1*). Approximately 8,6% (32) of genes upregulated in Rad21 RNAi were also induced when the transcriptional repressor Esg was knocked down (*Figure 6D and F*), *Figure 6—source data 1*), including the transcription factor nubbin/pdm1, known to be repressed by Esg (*Korzelius et al., 2014*). EC-specific genes including Jon and trypsin family proteases, and ser6, were also enriched (approximately 17%, 65 genes) after Rad21 knock down (*Dutta et al., 2015*; *Doupé et al., 2018*) (*Figure 6C and G*, *Figure 6—source data 1*). Note that we did not observe upregulation of cell death markers

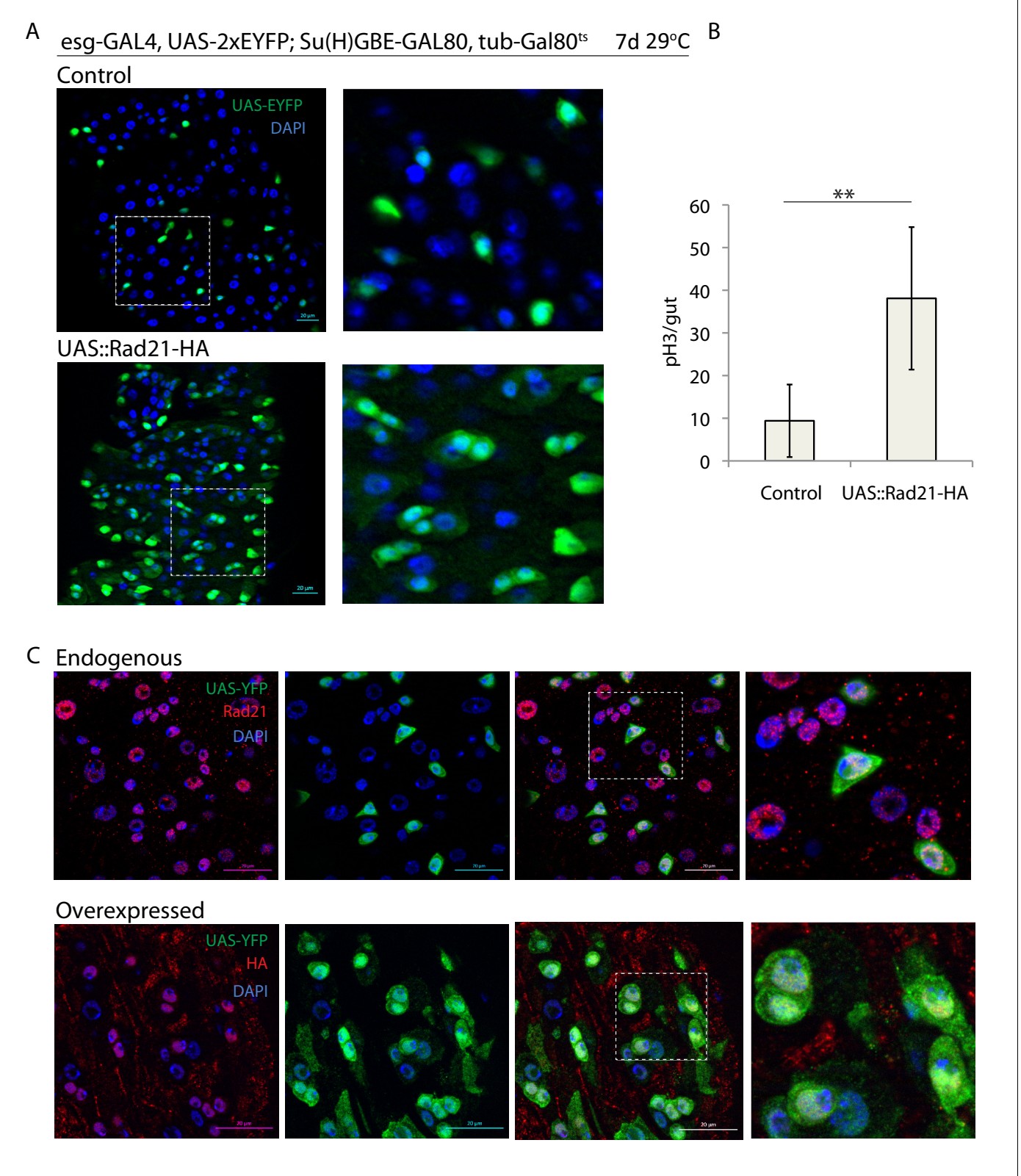

**Figure 3.** Rad21 overexpression leads to increased ISC proliferation. (**A**) esg^ts midguts expressing UAS-EYFP alone (control) or expressing UAS-Rad21-HA (**B**). Quantification of the number of mitotic pH3-positive cells/midgut in the guts from A (n = 8), (**C**). esg^ts midguts expressing UAS-EYFP alone (upper panels) or expressing UAS-Rad21-HA (lower panels). Samples were stained for GFP and either Rad21 (upper panels) or HA tag (lower panels). ANOVA. **p<0.01. Scale bars, 20 μm.

*Figure 3 continued on next page*

*Figure 3 continued*

The online version of this article includes the following figure supplement(s) for figure 3:

**Figure supplement 1.** Esg-FlipOut (F/O) midguts expressing UAS-EYFP alone (control) or expressing UAS-Rad21-HA.

after Rad21 knock down or over-expression (*Figure 6*, *Figure 6—source data 1*). These results were consistent with the differentiation phenotype of rad21-deficient ISCs (*Figure 1*), and led us to hypothesize that Rad21 may be required for Esg-mediated repression of differentiation genes. To test this hypothesis we performed a damID (*Korzelius et al., 2014*; *Marshall et al., 2016*) experiment to determine the genome-wide location of esg in wild-type and rad21 deficient ISCs. Using a previously described esg-dam construct, we observed 1034 significant peaks in esg-dam control samples, 862 of which were not observed in rad21 deficient ISCs, indicating that the vast majority of DNA interactions of esg in ISCs depend on functional rad21 (*Figure 6H Figure 7* and *Figure 6— source data 2*). GO analysis of peaks appearing in Esg-Dam in contrast to Esg-Dam/Rad21 RNAi yielded a number of genes involved in ISC maintenance and differentiation pathways (*Figure 6I and J*) anticipated due to the function of esg, the lost interaction sites included locations close to differentiation and EC specific genes, such as nubbin/pdm1 (*Figure 6K* and *Figure 6— source data 2*).

Finally, to verify our transcriptome data we performed qRT-PCR on FACS-purified Rad21-depleted ISCs and compared those to controls. We assessed expression of Pdm1 and esg, and found that, consistent with immunofluorescence data, Pdm1 was strongly induced after Rad21 downregulation, while Esg mRNA levels did not change after Rad21 knock down (*Figure 6—figure supplement 1*).

## Notch-independent stem cell differentiation in Rad21 loss-of-function conditions can be inhibited by esg over-expression

Altogether, our data indicated that Rad21 depletion results in induction of a transcriptional differentiation program. It remained unclear, however, how Rad21 interacts with canonical differentiation pathways in the ISC lineage. Since Notch signaling is the main pathway that initiates differentiation of ISCs into ECs (*Ohlstein and Spradling, 2007*; *Kapuria et al., 2012*; *Guo and Ohlstein, 2015*), we set to explore whether perturbing Notch signaling would interfere with ISC differentiation in Rad21 loss of function conditions. Loss of Notch in the ISC lineage perturbs differentiation, resulting in the formation of tumors consisting of ISCs and EEs. However, when Rad21 was knocked down in Notch-deficient ISCs, these cells still differentiated into Pdm1-positive cells (*Figure 8A and B*), and drastically inhibited ISC proliferation was observed (*Figure 8C*). ISC differentiation induced by loss of Rad21 is thus independent of Notch signaling.

Esg is a well-understood regulator of ISC maintenance that represses the expression of differentiation genes (*Korzelius et al., 2014*; *Loza-Coll et al., 2014*). The inhibition of Delta expression and the induction of Pdm1, both known Esg targets, after Rad21 downregulation, as well as our damID experiment, indicated a possible role for Rad21 in the regulation of the Esg transcriptional program. To functionally evaluate this hypothesis, we assessed whether Esg overexpression would influence differentiation of ISCs in Rad21 loss of function conditions. Esg overexpression robustly inhibited ISC differentiation in these conditions, significantly reducing the number of Pdm1-positive cells and increasing the frequency of mitotic ISCs in homeostatic as well as regenerative conditions (after infection) (*Figure 9*). These results indicate that Rad21 plays a critical role in the Esg-mediated maintenance of the stem cell state in the ISC lineage (*Figure 10*). At the same time, however, these data indicate that the requirement for Rad21 can be overcome by elevating Esg expression levels, suggesting that Rad21 plays mostly an accessory role to ensure robust Esg-mediated gene regulation.

## Discussion

Our data support a specific function for Cohesin-mediated chromatin regulation in the control of somatic stem cell biology. We find that both Rad21 gene function, as well as the activity of Nipped-B and Polo, previously described as regulators of Rad21 chromosome association, influence stem cell

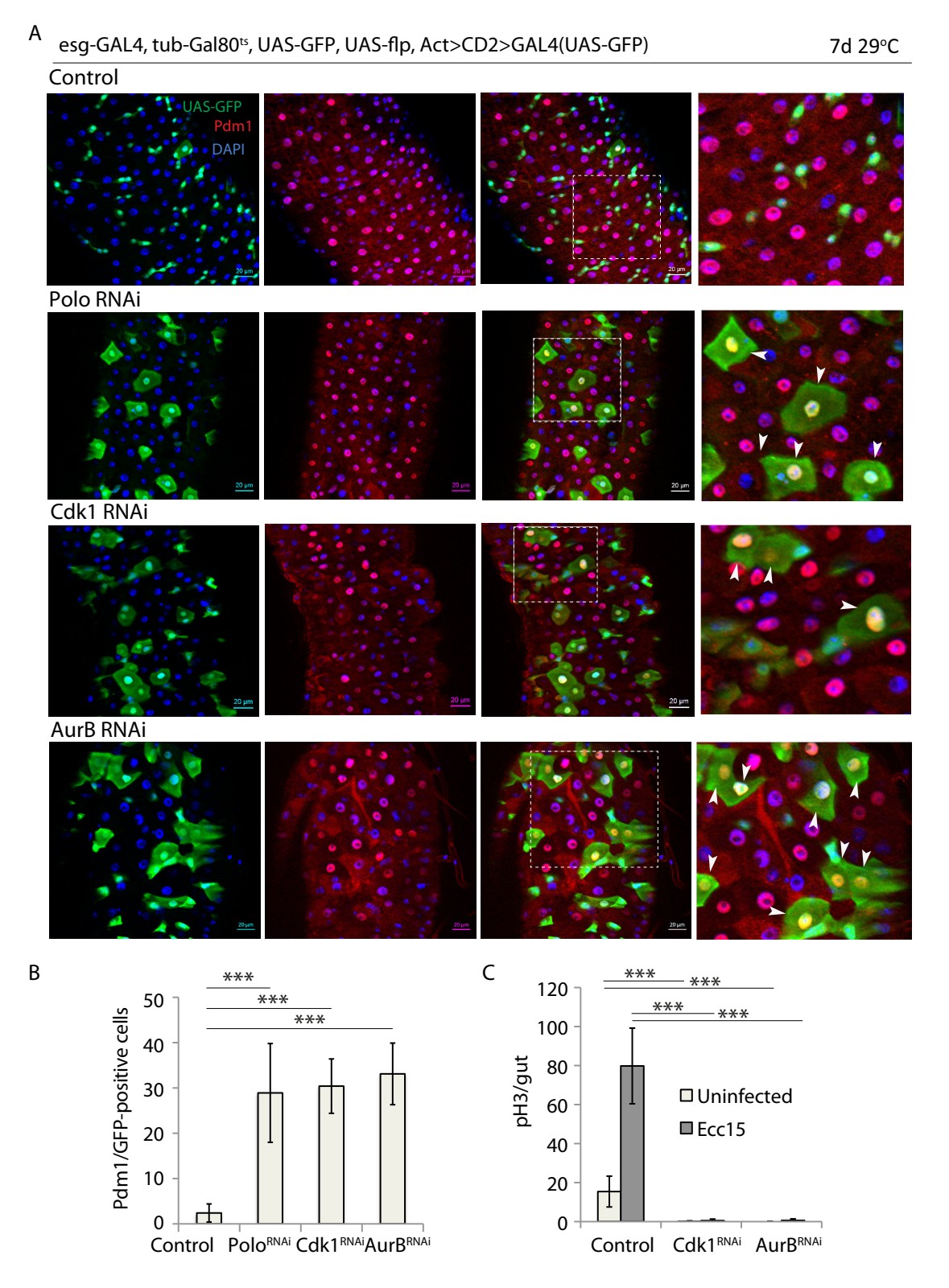

**Figure 4.** Knock down of cell cycle regulators leads to premature ISC differentiation. (**A**) esg-F/O midguts expressing UAS-GFP alone (control) or expressing polo[RNAi]. cdk1[RNAi] and aurB[RNAi]. Samples were stained for GFP and Pdm1. (**B**) Quantification of GFP-positive/Pdm1-positive cells from A. (n = 609, 369, 500, 482). (**C**) Quantification of the number of mitotic pH3-positive cells/midgut in esg[ts] guts expressing UAS-GFP alone (control) or

*Figure 4 continued on next page*

*Figure 4 continued*

cdk1$^{RNAi}$ and aurB$^{RNAi}$ with and without Ecc15 infection (n = 7–11), ANOVA. \*\*\*p<0.001. Differentiated cells are labeled with white arrowheads. Scale bars, 20 μm.

maintenance in the *Drosophila* ISC lineage. Our observations support the idea that the Cohesin complex interacts with interphase chromatin and supports selected regulators of gene expression.

It has been proposed that the 3-dimensional genome structure is more plastic in stem cells compared to differentiated progeny, thus facilitating re-shaping into the more stable chromatin organization that is critical for the establishment of a fully differentiated daughter cell with a specific transcriptional program (*Filion et al., 2010*; *Sexton and Cavalli, 2015*). According to this model, maintaining plasticity of genome organization is thus critical to maintain stemness and prevent premature differentiation. Our findings, together with other recent studies, indicate that Cohesin plays a central role in maintaining this plasticity. At the same time, these studies highlight the selectivity and cell-type specificity of Cohesin function in different stem cell lineages: Cohesin depletion increases differentiation of skin stem cells, but inhibits hematopoietic stem cell (HSC) differentiation (*Mullenders et al., 2015*; *Galeev et al., 2016*; *Noutsou et al., 2017*; *Cuartero et al., 2018*). In HSCs, Cohesin mediates recruitment of the transcription factor NFκB to chromatin and induces immune and pro-survival gene expression (*Mullenders et al., 2015*; *Galeev et al., 2016*; *Cuartero et al., 2018*). It has further been reported to control chromatin accessibility and activity of ERG, GATA2 and RUNX1 transcription factors in HSCs (*Mazumdar et al., 2015*). During hematopoiesis, the Polycomb protein ASLX1 interacts with Cohesin to maintain chromatid separation and transcription (*Li et al., 2017*). By mediating recruitment and transcriptional activity of the pluripotency factors Klf4, Esrrb and Sox2, Cohesin also regulates pluripotency of embryonic stem cells (*Nitzsche et al., 2011*).

In the mammalian intestine, elevated Rad21 expression has been implicated in loss of heterozygosity at the APC locus, as well as the activation of L1 retrotransposons (*Xu et al., 2014*). This effect suggests that Rad21 can also act as a tumor-promoting factor in the intestinal epithelium. This study did not observe defects in stem cell self-renewal efficiency in Rad21 heterozygotes, but did observe reduced expression of the ISC marker Lgr5 in heterozygous crypts. ISCs with homozygous Rad21 loss of function were not examined, however, and our results indicate that detailed lineage tracing of Rad21 mutant ISCs would be of interest in the mammalian ISC lineage.

Our work identifies Esg-mediated gene expression as the primary target of Cohesin in ISCs. Esg negatively regulates expression of differentiation factors such as Pdm1 and activates stemness genes including Delta and cell cycle proteins (*Korzelius et al., 2014*; *Loza-Coll et al., 2014*). Loss of Rad21 phenocopies the transcriptional consequences of Esg loss in ISCs, and it is striking that Esg overexpression is able to prevent ectopic ISC differentiation in the Rad21 loss of function background. This suggests that excess levels of Esg protein can exert the required repressive effect on differentiation genes even in the absence of Rad21-mediated genome re-organization. We propose that Cohesin supports Esg function by facilitating access of the transcription factor to promotor regions of differentiation genes, and that in the absence of cohesion, chromatin organization is relaxed, allowing ectopic expression of differentiation genes (*Figure 10*). By providing excess Esg protein, such promoters can then be repressed even in the absence of Rad21. Similar consequences of Cohesin perturbations have been reported in other cell types (*Pauli et al., 2010*; *Dorsett and Ström, 2012*; *Zuin et al., 2014*; *Lupiáñez et al., 2015*; *Rao et al., 2017*; *Schwarzer et al., 2017*).

Mechanistically, how may Cohesin regulate Esg-mediated gene expression? In other systems, Cohesin has been shown to regulate gene expression via interaction with the Mediator complex of Polymerase II (*Kagey et al., 2010*). Such a function of Cohesin in the transcriptional regulation complex of Esg is supported by the fact that depletion of Mediator subunits in ISCs can cause premature differentiation (*Zeng et al., 2015*), reminiscent of the Cohesin loss of function phenotype. Our damID data indicate that in the absence of rad21, esg loses its interaction with locations close to its target genes, but additional work is needed to dissect other mechanistic aspects of Cohesin-mediated transcriptional regulation in ISCs, such as transcription factor recruitment to chromatin, modifications and changes in chromatin conformation that lead to the transcriptional changes described here, and that ensure stem cell maintenance.

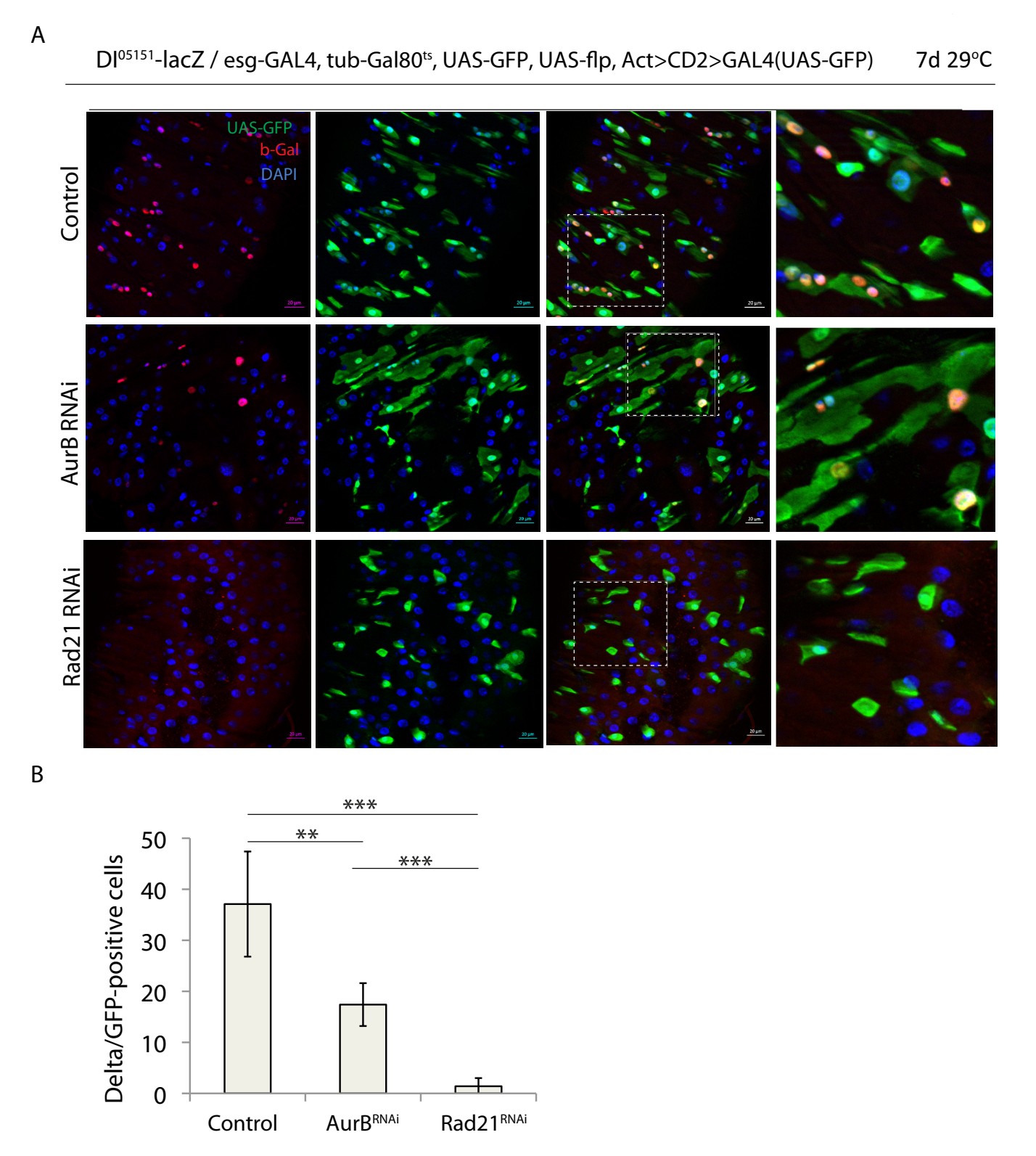

**Figure 5.** Downregulation of AurB and Rad21 differentially affect ISC maintenance. (**A**) esg-F/O/Delta-lacZ midguts expressing UAS-GFP alone (control), aurB[RNAi] or expressing rad21[RNAi]. Samples were stained for GFP and b-gal. (**B**) Quantification of GFP-positive/Delta-lacZ-positive cells from A. (n = 1489, 1522, 733), ANOVA. **p<0.01, ***p<0.001. Scale bars, 20 μm.

*Figure 5 continued on next page*

*Figure 5 continued*

The online version of this article includes the following figure supplement(s) for figure 5:

**Figure supplement 1.** Rad21 intenstiy is reduced in Polo$^{T182D}$-expressing ISCs.

Aneuploidy has been recently linked to stem cell maintenance and proliferation (*Gogendeau et al., 2015*; *Resende et al., 2018*). Based on our findings and our observations, we propose that loss of Cohesin results in ISC differentiation independently of aneuploidy for a number of reasons: First, we find that both depleting Cohesin as well as promoting its release from DNA induces ISC differentiation. The various perturbations to that effect (two independent Rad21 shRNAs, Nipped B shRNA, T182D Polo mutant) elicited a significantly stronger differentiation phenotype compared to perturbations of other regulators of mitosis (AurB, Cdk1, Polo), indicating that the mitotic role of Cohesin could not completely account for the observed robust differentiation of ISCs into ECs (*Figure 1*, *Figure 2* and *Figure 3*). Importantly, all of these perturbations resulted in a complete inhibition of proliferation, indicating a comparable level of cell cycle arrest (*Figure 1D*, *Figure 2—figure supplement 1* and *Figure 4C*). Second, the fact that Nipped B knockdown also results in ISC differentiation strongly supports an aneuploidy-independent mechanism, as the binding of Cohesin to chromatin after completion of mitosis is mediated in interphase by the complex containing Nipped B. These data thus suggest that Cohesin binding to chromatin in interphase is critical for ISC maintenance (*Figure 2A and C*). Similar important roles of Cohesin in interphase cells have been recently described in mammals (*Meisenberg et al., 2019*). Third, ISCs are normally quiescent in unchallenged young fly midguts and enter mitosis at a very low rate (up to approximately 10–20%) (*Figure 1D*, *Figure 2—figure supplement 1*, *Figure 3B* and *Figure 4C*). Cohesin depletion in ISCs would thus only lead to aneuploidy in the small subset of ISCs that enter mitosis. In contrast, Cohesin depletion led to differentiation of around 60–70% of all ISCs (*Figure 1B*), and to an almost complete elimination of Delta-positive cells (to approximately 2%) (*Figure 5B*). Such a drastic effect does not reflect the frequency of cells entering mitosis. Finally, our RNAseq and DamID analyses suggest that Cohesin has profound impact on the transcriptome of ISCs, and significantly affects the transcriptional program regulated by Escargot (*Figure 6*). Accordingly, Cohesin depletion led to loss of Esg promotor binding, and significantly reduced Esg-regulated transcripts. Esg over-expression in Cohesin-depleted cells further significantly rescued the premature ISC differentiation (*Figure 9*) and restored homeostasis. It seems unlikely that these effects on Esg promoter loading and Esg-mediated gene regulation are merely an indirect consequence of aneuploidy. It is important to point out further that differentiation as a consequence of aneuploidy does not seem to be a robust phenotype: While depletion of *bub3* was found to induce ISC differentiation and thus loss of ISCs, depletion of BubR1, mad2, or mps1 all result in increased ISC proliferation and an accumulation of ISCs/EBs and EEs in the intestinal epithelium (*Gogendeau et al., 2015*; *Resende et al., 2018*).

We anticipated that downregulation of the chromatin insulator CTCF might recapitulate the Rad21 RNAi phenotype but, on the contrary, we have not observed any detectable rise in ISC differentiation (*Figure 2*). Several recent studies on CTCF in *Drosophila* found little evidence of CTCF in chromosome domain/loop formation (*Ramírez et al., 2018*; *Wang et al., 2018*; *Matthews and White, 2019*). Other insulator protein complexes such as BEAF-32/CP190 or BEAF-32/Chromator have been implicated in TAD organization in *Drosophila* (*Ramírez et al., 2018*; *Wang et al., 2018*; *Matthews and White, 2019*). Therefore, it has been recently proposed that fly CTCF is not a major boundary definition protein but is rather a regulator of Hox or other gene expression (*Gambetta and Furlong, 2018*; *Szabo et al., 2019*).

Cohesin has been suggested to maintain transcriptional memory after mitosis (*Yan et al., 2013*). Contrary to original observations of complete Cohesin removal, some level of Cohesin has been detected in mitosis and suggested to mediate reloading of transcription factors onto DNA to resume gene expression in interphase (*Yan et al., 2013*) or maintain gene expression in mitosis (*Teves et al., 2016*). This would be especially relevant for stem cells highly dependent on persistent expression of stemness genes (*Ferraro et al., 2016*). Based on these reports and our data, it is possible that Rad21 in ISCs ensures reloading of Esg onto relevant promoters after mitosis to maintain stable expression of stemness genes and ensure self-renewal (*Gogendeau et al., 2015*) (*Figure 10*). Accordingly, we show that downregulation of mitotic factors and Rad21 release from chromatin

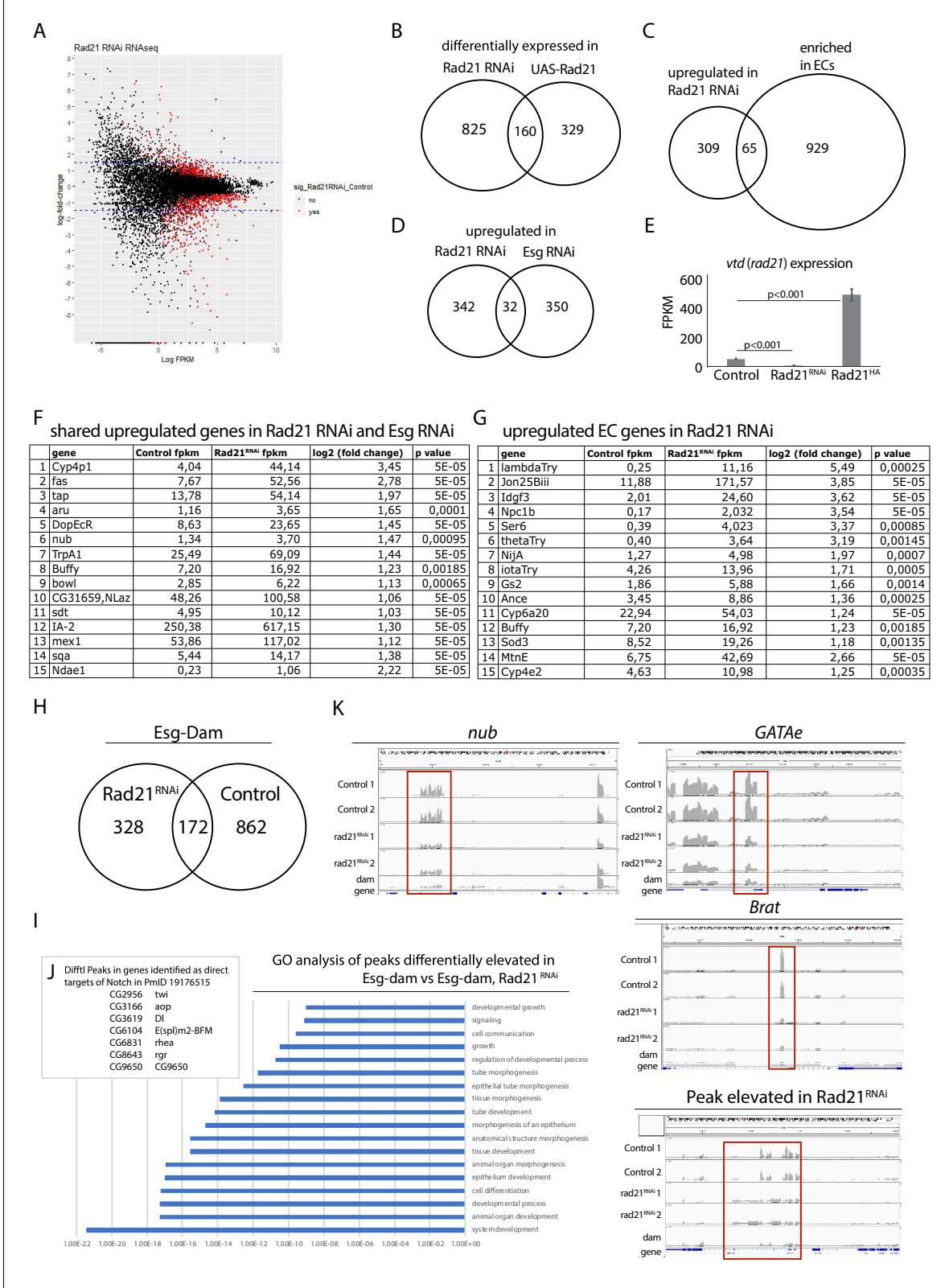

**Figure 6.** RNAseq and DamID analysis of ISCs after downregulating or overexpressing Rad21. (**A**) Transcriptomic analysis of ISCs expressing UAS-EYFP (Control) or Rad21$^{RNAi}$ or UAS-Rad21-HA. Scatter plot of gene expression regulation after Rad21 knock down. Red dots represent significant hits (false discovery rate q < 0.05). (**B**) Overlap between significantly differentially expressed genes found in Rad21 RNAi and UAS-Rad21-HA datasets, Fisher exact test (***p<0.001). (**C**) Overlap between genes upregulated in Rad21 RNAi dataset and EC-specific genes (**Doupé et al., 2018**), Fisher exact test

*Figure 6 continued on next page*

*Figure 6 continued*

(***p<0.001). (**D**) Overlap between genes upregulated in Rad21 RNAi and esg RNAi datasets, Fisher exact test (***p<0.001) (*Korzelius et al., 2014*). (**E**) Rad21 expression in Control, Rad21$^{RNAi}$ or UAS-Rad21-HA ISCs from (**A**). (**F**) Examples of EC genes upregulated after Rad21 knock down from C. (**G**) Examples of genes shared in Rad21 RNAi and esg RNAi datasets from D. (**H**) DamID analysis of ISCs expressing Esg-Dam (Control) and Esg-Dam/Rad21$^{RNAi}$ (Rad21 RNAi). Overlap between significant peaks found in Esg-Dam compared to Dam only and significant peaks in Esg-Dam/Rad21$^{RNAi}$ compared to Dam only. (**I**) GO analysis of peaks significantly elevated in Esg-Dam compared to Esg-Dam/Rad21 RNAi samples. (**J**) Differential peaks as direct targets of Notch from H. (**K**) Differential occupancy of nub, GATAe and Brat loci by Esg-Dam or Esg-Dam after Rad21 knock down. Dam only serves as control. Lower panel shows peaks elevated after Rad21 knock down.

The online version of this article includes the following source data and figure supplement(s) for figure 6:

**Source data 1.** Transcriptomic analysis of ISCs expressing UAS-EYFP (Control) or Rad21$^{RNAi}$ or UAS-Rad21-HA.

**Source data 2.** DamID analysis of ISCs expressing Esg-Dam in a wild-type (control) or Rad21 deficient (Rad21-) background.

**Figure supplement 1.** qRT-PCR analysis of Pdm1 and Esg expression in ISCs after Rad21 knock down, ANOVA.

leads to premature ISC differentiation. Furthermore, expression of constitutively active Polo, which is known to force cells into mitosis (*Sumara et al., 2002*; *van de Weerdt et al., 2005*), remove Cohesin from chromatin, and extend the duration of mitosis, triggers drastic ISC differentiation.

Recent evidence indicates that chromatin loops to a great extent could be reestablished shortly after Cohesin removal from chromatin (*Rao et al., 2017*). This could imply other mechanisms controlling chromatin organization and therefore regulating stem cell transcription and 'bookmarking' transcription sites in mitosis. Alternatively, this may suggest that Cohesin downregulation could lead to chromatin loop rearrangements even though partially preserving chromatin complexity but still switching transcriptional programs of stem cells towards differentiated progeny. It is also interesting that acute Cohesin depletion slightly inhibits but nevertheless preserves cell proliferation of developing neuroblasts (*Mirkovic et al., 2019*) indicating a complex and still understudied role of Cohesin in mitosis, which may include cell type-, stage- or organism-dependent differences in mitosis, involve other cellular processes and non-cell autonomous mechanisms. All together, our work supports the notion that characterizing chromatin dynamics in somatic stem cell lineages, and dissecting the multiple roles of Cohesin in stem cell maintenance, mitosis, and differentiation in a variety of stem cell lineages and organisms, is critical to understand somatic stem cell regulation and dysfunction.

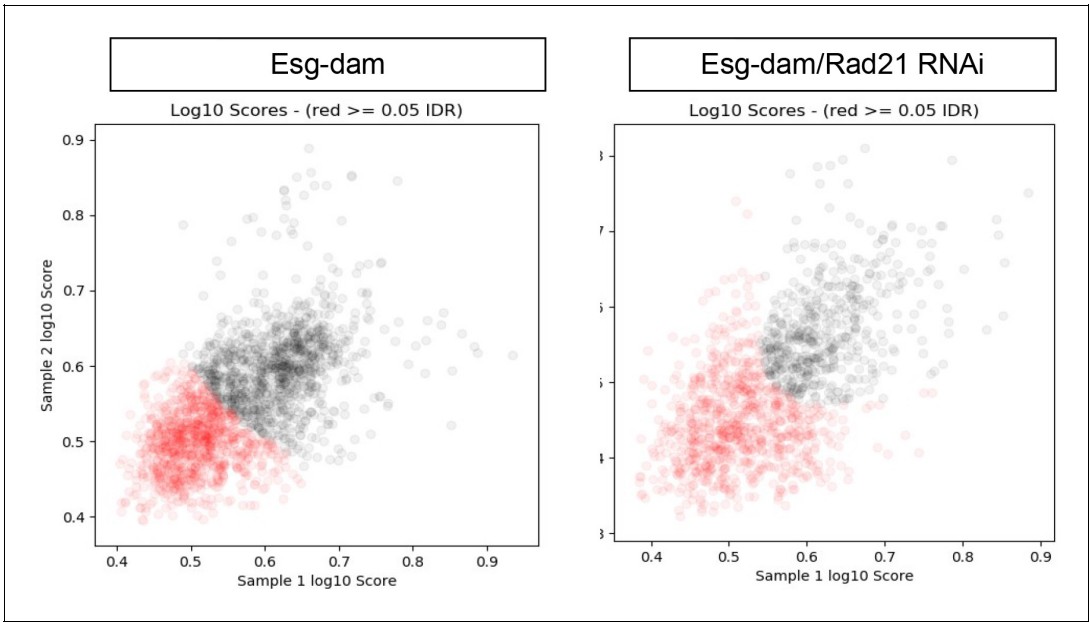

**Figure 7.** DamID analysis of ISCs expressing Esg-Dam (left) or Esg-Dam with Rad21$^{RNAi}$ (right). Scatter plots showing reproducible peaks in repeat samples identified by the idr_tools pipeline. Black dots represent peaks with higher reproducibility (IDR <0.05).

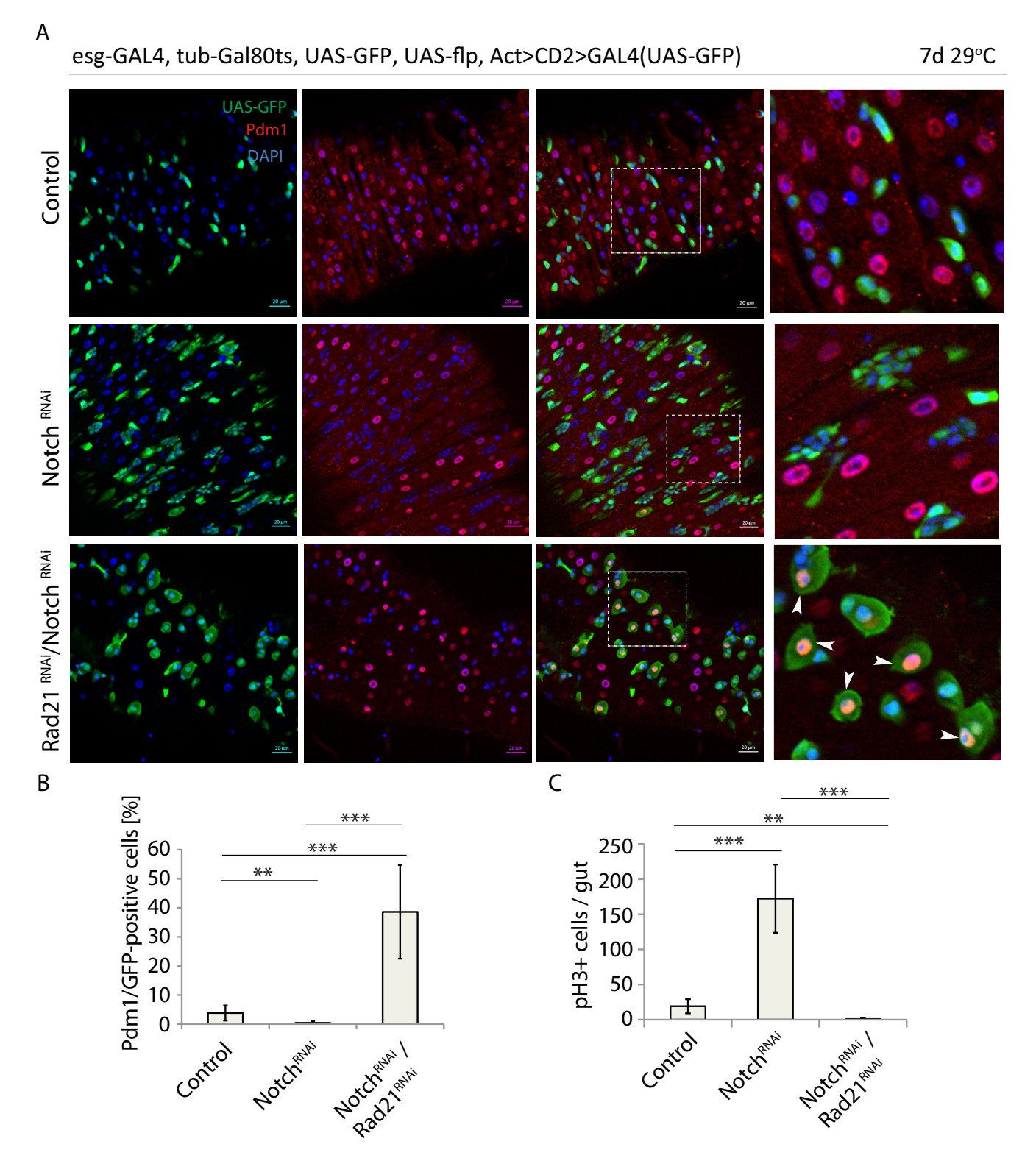

**Figure 8.** Rad21 RNAi-induced ISC differentiation occurs independently of Notch signaling. (**A**) esg-F/O midguts expressing UAS-GFP alone (control), notch$^{RNAi}$ or notch$^{RNAi}$/rad21$^{RNAi}$. Samples were stained for GFP and Pdm1. (**B**) Quantification of GFP-positive/Pdm1-positive cells from A. (n = 829, 2485, 376). (**C**) Quantification of the number of mitotic pH3-positive cells/midgut in esg$^{ts}$ guts expressing UAS-GFP alone (control), notch$^{RNAi}$ or

*Figure 8 continued on next page*

*Figure 8 continued*

notch^RNAi/rad21^RNAi with and without Ecc15 infection (n = 6–7), ANOVA. **p<0.01, ***p<0.001. Differentiated cells are labeled with white arrowheads. Scale bars, 20 μm.

## Materials and methods

### Fly stocks

The following stocks study were reported previously (*Korzelius et al., 2014*).

Esg^ts driver: y,w;esg-GAL4/CyO;tub-GAL80ts,UAS-GFP/Tm6B, Esg-Flip-Out (F/O) driver lines: w; esg-Gal4,UAS-GFP,tub-Gal80ts/CyO;UAS-flp,act >CD2>Gal4/Tm6B and w;esg-gal4, tub-Gal80ts, UAS-GFP/CyO,wg-lacZ;P{w[+mC]=UAS FLP.D}JD2/Tm6B

Esg overexpression: y,w;UASt::esg/CyO

Rad21 overexpression: w;UAS-Rad21-HA/Cyo (a gift from Stefan Heidmann, Bayreuth and Raquel Oliviera, Gulbenkian).

Delta reporter: w;If/CyO,wglacZ; Dl05151-lacZ/Tm6B

DamID lines:

Dam only: w;M{w[+mC]=hs .min(FRT.STOP1)dam}ZH-51C/CyO,wg-lacZ

Esg-Dam: w;esg-Dam(ZH-51C) M4M1/CyO, wg-lacZ;MKRS/Tm6B

Notch knock down: UAS-Notch^RNAi.

The following *Drosophila* lines were obtained from Bloomington Stock Collection (Indiana): Rad21 RNAi (#36786, #65229), Cdk1 RNAi (#28368), AurB RNAi (#58308), Polo T182D (#8434), Polo RNAi (#35146, #35770), Nipped-B (#32406, #36614), CTCF RNAi (#40850).

Flies containing Gal80ts were raised at 18°C and used at the minimum of 3 days of age. Animals were then shifted to 29°C for 7 days unless indicated.

### Infection with *Erwinia carotovora*

Bacteria were inoculated and grown overnight at 30 degrees, then collect and resuspended in 5% sucrose in water. Flies were shifted to 29°C for 7 days unless indicated to induce RNAi or transgene expression. Flies were then first starved for 4 hr in vials with water and subsequently flipped in new vials with Ecc15. Infections were conducted for 14–16 hr unless otherwise indicated. Esch experiment was conducted twice.

### Antibodies

Chicken anti-GFP (A10262, Invitrogen), mouse monoclonal anti-GFP (sc-9996, B-2, Santa-Cruz) rabbit polyclonal anti-Pdm1 (kindly provided by Cai Yu, Singapore), rabbit polyclonal anti-Rad21/vtd (kindly provided by Dale Dorsett, Saint Louis), rabbit polyclonal anti-pH3 (sc-8656, Santa Cruz), mouse monoclonal anti- beta-galactosidase/LacZ (40-1a, DSHB), HA-tag mouse monoclonal antibody (HA-7, H3663, Sigma). Secondary antibodies: goat polyclonal anti-chicken, rabbit, mouse Alexa 488, Alexa 568-coupled (A11039, A11036, A11034, A11029, A11031, Invitrogen). DAPI was used to stain DNA.

### Immunofluorescence

Protocol was adapted from the study described previously (*Resnik-Docampo et al., 2017*). Intestines from adult female flies were dissected into phosphate buffered saline (PBS) and fixed in in 4% paraformaldehyde (PFA) for 60 min at room temperature in nutating mixer. Guts were pemeabilized and washed by sequentially incubating in 0.5% Triton-x100 (Tx-100)/PBS, 0.5% Na-Deoxycholate (NaDoc), 0.3% Tx-100 for 10 min and blocked in 0.3% Tx-100/0.5 BSA/PBS for 30 min. Primary antibody in blocking solution was then added to guts and incubated overnight. Gut were further washed 4 times in 0.3% Tx-100 for 10 min. Then secondary antibody diluted in blocking solution was added, and guts were incubated for 2 hr. Afterward guts were washed 3 times in 0.3% Tx-100 for 10 min and incubated in DAPI for 5 min. Gut were subsequently washed once in 0.3% Tx-100 for 10 min and mounted in Vectashield (Vector Laboratories). Images were taken using Axiovert equipped with Apotome V or LSM710 confocal microscope and further analysed using Aviovision (Zeiss) and Image J software.

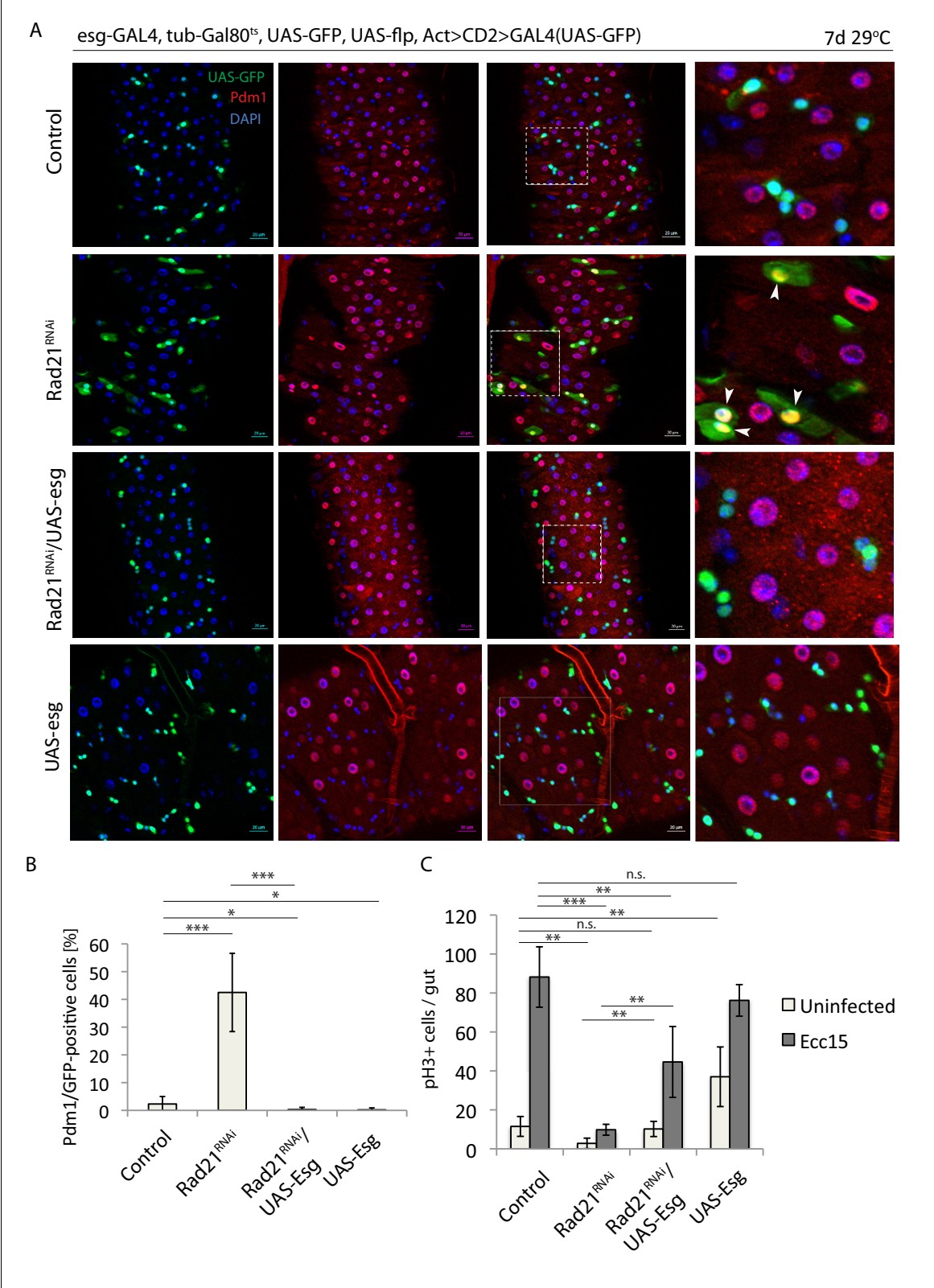

**Figure 9.** Escargot overexpression prevents Rad21 RNAi-induced ISC differentiation. (**A**) esg-F/O midguts expressing UAS-GFP alone (control), expressing rad21$^{RNAi}$, UAS-esg $^{RNAi}$/rad21$^{RNAi}$ or UAS-esg. Samples were stained for GFP and Pdm1. (**B**) Quantification of GFP-positive/Pdm1-positive cells from A. (n = 491, 402, 391, 766), (**C**) Quantification of the number of mitotic pH3-positive cells/midgut in esg$^{ts}$ guts expressing UAS-GFP alone
*Figure 9 continued on next page*

*Figure 9 continued*

(control), rad21[RNAi], UAS-esg/rad21[RNAi] or UAS-esg with and without Ecc15 infection (n = 4–6), ANOVA. *p<0.05, **p<0.01, ***p<0.001, n.s. not significant. Differentiated cells are labeled with white arrowheads. Scale bars, 20 µm.

### Quantification of pH3, Pdm1, DAPI, RAD21 and *Delta signals*

Images were taken using Apotome (Zeiss). Mitotic index in fly guts was scored by pH3-positive ISCs in esg[ts] crosses. Mounted guts were inspected under microscope and pH3-positive ISCs (labeled by UAS-GFP) were counted. 6–12 guts per sample were scored in each experiment; each experiment was performed 2–3 times (biological replicas). Pdm1 signals were quantified in posterior midguts from flies derived from EsgF/O crosses by Axiovision software (Zeiss). Pdm1 positive ISCs (labeled with UAS-GFP) were scored in controls and in RNAi midguts. Images from 8 to 10 guts within 2–3 experiments (biological replicas) were analysed and results were pulled and presented. Delta-positive ISCs were quantified as described for Pdm1. Size of DAPI-positive nuclei and Rad21 intensity were measured in posterior midguts using Image J. A line around each of 50 DAPI and Rad21-positive nuclei was drawn and either size or average pixel intensity were analysed using Image J.

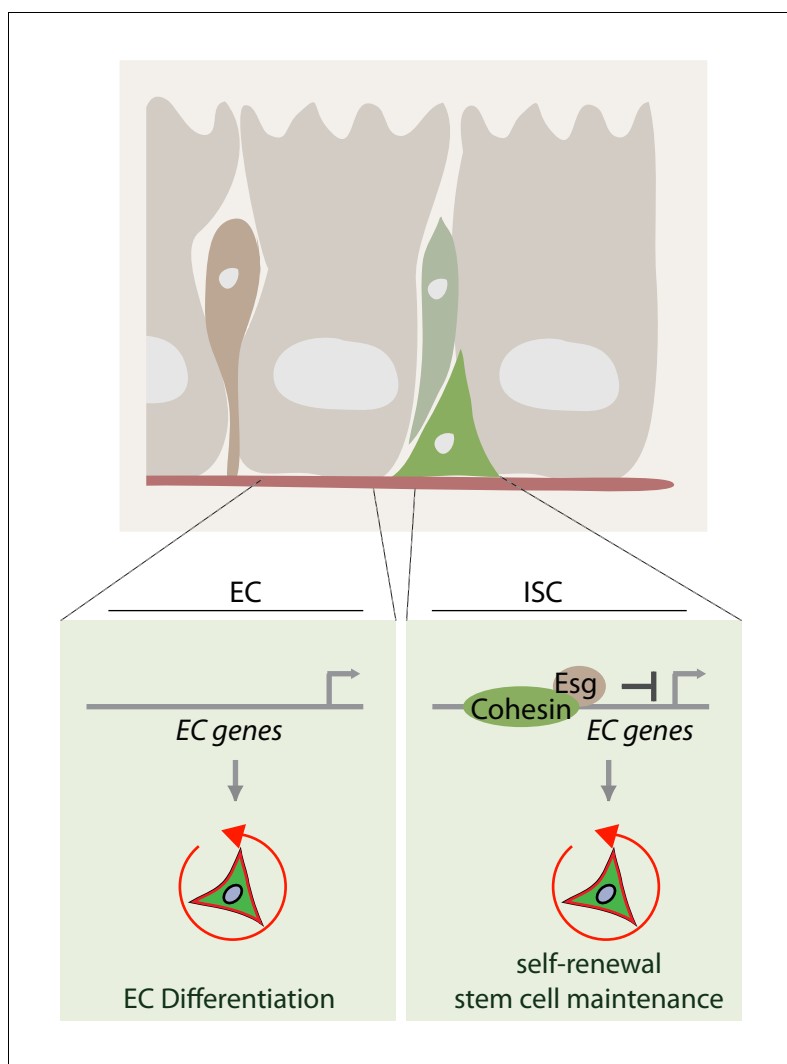

**Figure 10.** Model of how rad21 maintains ISC identity and proliferation via transcription factor esg. Rad21-containing Cohesin complex recruits esg to promotors of gene responsible for ISC maintenance and repression of differentiation (for example Pdm1). Rad21 loss triggers EC gene expression and ISC differentiation.

## RNA isolation

Intestines (50/sample) in four independent experiments were dissected in PBD and homogenized by incubating in elastase for 1 hr at 27 degrees with vigorous shaking and pipetting. Cell suspensions were then used for FACS sorting (FACS Aria) and gated based on UAS-GFP signal set by w1118 cells autofluorescence. 10.000–20.000 GFP-positive cells were then processed for RNA isolation using Arcturus PicoPure RNA Isolation Kit (KIT0202, KIT0204, Thermo) according to manufacturer's instructions. RNA was subsequently used for cDNA preparation using QuantiTect-Reverse Transcription Kit (205311, Qiagen) or RNAseq.

## qRT-PCR

PCR was performed 3 times using Thermo Scientific Maxima SYBR Green/ROX qPCR Master Mix (#K0222) according to manufacturer's instructions. Pdm1 and Esg mRNA levels in rad21[RNAi] samples were normalized to housekeeping genes Gdh and Rpl32 and presented as ratio to controls (UAS-GFP). Primers: Gdh F: 5'-gctccgggaaaaggaaaa-3', R: 5-tccgttaattccgatcttcg-3'; RpL32 F: 5'-atcgtgaa-gaagcgcaccaa-3', R: 5'-tgtcgataccccttgggcttg-3'; Esg F: 5'-cgccagacaatcaatcgtaagc-3', R: 5'-tgtgtacgcgaaaaagtagtgg-3'; Pdm1 (Nub) F: 5'-cgggataaatcgaaggaagc-3', R: 5'-agtatttgatgtgtttgc-gacttt-3'.

## RNAseq

RNA-seq dual-indexed TrueSeq stranded mRNA kit was used to prepare RNA-seq libraries. Libraries sequenced on Illumina Hiseq machine using two different lanes. Read length is 51 bases and number of reads varied across samples from 8.7M reads to 64.1M reads. First we used *Trimmomatic* tool version 0.36 (*Bolger et al., 2014*) to trim and filter low quality reads. Then, we used *Tophat2* version 2.1.1 (*Kim et al., 2013*) to align reads that passed quality control to the UCSC dam6 reference genome. The vast majority of the reads passed quality control and the overall read mapping rate is consistently between 96.7% and 97.4% across samples. To normalize between read counts and to find differentially expressed genes between Rad21RNAi, Rad21HA, and control we used *Cufflink* suite of tools version 2.2.1 (*Trapnell et al., 2010*; *Roberts et al., 2011b*; *Roberts et al., 2011a*; *Trapnell et al., 2013*).

## Statistics

Data are presented as mean SD or SEM. Statistical analysis of two experimental groups was performed using one-way ANOVA test. For non-parametric distributed data, the Mann-Whitney-U-test or Fisher exact test was applied. Significance was considered at $p < 0.05$. *$p < 0.05$, **$p < 0.01$, ***$p < 0.001$.

## DamID

Protocol was adapted from the study described previously (*Marshall et al., 2016*). Intestines (10–20/sample) in two independent experiments were dissected in PBD and used for extraction of genomic DNA using QIAamp DNA Micro Kit (56304, Qiagen). DNA was further digested by DpnI and cleaned

**Table 1.** Dam ID alignment.
Number of reads is uniform across samples, the sequence quality is good and uniform. Alignment rate is good and uniform across dam and Esg-Dam samples but significantly lower for the Esg-Dam/Rad21 samples.

|  | # reads | #reads containing no Ns | Overall alignment rate |
|---|---|---|---|
| dam1 | 62,324,161 | 99.88 | 94.46% |
| dam2 | 61,289,068 | 99.88 | 95.17% |
| esg1 | 63,277,266 | 99.88 | 92.87% |
| esg2 | 64,279,981 | 99.88 | 92.41% |
| rad21esg1 | 66,375,925 | 99.88 | 56.21% |
| rad21esg2 | 63,063,034 | 99.88 | 79.86% |

**Table 2.** DamID peak calling.

|  | #low threshold peaks | #merged peaks | #peaks passing IDR cutoff of 0.05 | %peaks passing IDR cutoff of 0.05 |
|---|---|---|---|---|
| esg-dam1 | 2521 | 1973 | 1034 | 52.4% |
| esg-dam2 | 2537 | | | |
| rad21RNAi/esg-dam1 | 2387 | 1327 | 500 | 37.7% |
| rad21RNAi/esg-dam2 | 2175 | | | |

via PCR purification. Digested DNA fragment were used for ligation of DamID adaptors and digestion by DpnII and PCR amplification. DNA was futher purified and sonicated to reduce fragment size. DNA quality was controlled by agarose gel and Agilent DNA Bioanalyser system. Samples were then used for library preparation and sequencing. NGS library was prepared using Illumina TruSeq nano 529 DNA kit. After library quality control sample were subsequently sequenced as 50 bp 530 single-end on an Illumina HiSeq2500.

### DamID data analysis

Read alignment and peak calling: We used *damid_pipeline* script from Marshall OJ's *damid_pipeline* software version 1.4.4 (*Marshall and Brand, 2015*) to align fastq files to the *Drosophila* dm6 reference genome and to produce coverage files. Alignment rate was good and uniform across Dam control and Esg-Dam samples but was significantly lower for the Esg-Dam(rad21Δ) samples as shown in *Table 1*. Then, to identify DNA regions with higher mapping counts in Esg-Dam samples compared to a Dam-control sample we used *find_peaks* script from the same software. *find_peaks* was called with the option –fdr = 0.1, allowing peaks with relatively high level of false discovery rate. The pipeline was set to compare between samples according to *Table 2*. Reproducibility analysis: We used the *idr* tool version 2.02 to merge reproducible peaks in both replicates of Esg-Dam and two replicates of Esg-Dam(Rad21Δ). Gene annotation: To find genes in proximity to genomic regions identifies as reproducible peaks, we used the BioConductor *annotatr* R package version 1.8.0 (*Cavalcante and Sartor, 2017*). Differential binding analysis: To identify differential targets of esg between control and Rad21Δ backgrounds we first used *mergePeaks* tool from the *HOMER* software (*Heinz et al., 2010*) to merge peaks. We merged peaks that passed IDR analysis in Esg-Dam samples with peaks that passed IDR analysis in Esg-Dam(Rad21Δ) samples. *mergePeak*s resulted in a list of 1362 peaks. Then we used *HOMER*'s *getDifferentialPeaks* tool to find 902 peaks that are enriched in Esg-Dam compared to Esg-Dam(Rad21Δ) and vice-versa, 328 peaks that are enriched in Esg-Dam (Rad21Δ) compared to Esg-Dam. Cutoff used for enrichment was fold-change of 1.2 or greater.

## Acknowledgements

We are indebted to Dale Dorsett, Stefan Heidmann, Raquel Oliviera and Cai Yu for reagents. We especially thank Jerome Korzelius for assistance in planning and executing experiments and for providing fly stocks. We also acknowledge support from the FLI core facilities Sequencing, FACS and Imaging. Work at the Buck Institute was funded by NIH R01 GM117412.

## Additional information

### Competing interests

Heinrich Jasper: employee of Genentech, Inc, a member of the Roche group. The other authors declare that no competing interests exist.

### Funding

| Funder | Grant reference number | Author |
|---|---|---|
| National Institutes of Health | NIH R01 GM117412 | Heinrich Jasper<br>Tal Ronnen-Oron |

The funders had no role in study design, data collection and interpretation, or the decision to submit the work for publication.

## Author contributions

Aliaksandr Khaminets, Conceptualization, Data curation, Software, Formal analysis, Supervision, Validation, Investigation, Visualization, Methodology, Project administration; Tal Ronnen-Oron, Resources, Data curation, Software, Formal analysis, Investigation, Visualization, Methodology; Maik Baldauf, Elke Meier, Resources, Data curation, Investigation, Methodology; Heinrich Jasper, Conceptualization, Data curation, Software, Supervision, Funding acquisition, Validation, Visualization, Methodology, Project administration

## Author ORCIDs

Aliaksandr Khaminets ⬚ https://orcid.org/0000-0001-9143-8312
Heinrich Jasper ⬚ https://orcid.org/0000-0002-6014-4343

## Decision letter and Author response

Decision letter https://doi.org/10.7554/eLife.48160.sa1
Author response https://doi.org/10.7554/eLife.48160.sa2

## Additional files

### Supplementary files

• Transparent reporting form

### Data availability

Datasets generated during this study is included in Source data files.

The following previously published datasets were used:

| Author(s) | Year | Dataset title | Dataset URL | Database and Identifier |
|---|---|---|---|---|
| David P Doupé, Owen J Marshall, Hannah Dayton, Andrea H Brand, Norbert Perrimon | 2018 | Drosophila intestinal stem and progenitor cells are major sources and regulators of homeostatic niche signals | https://www.ncbi.nlm.nih.gov/geo/query/acc.cgi?acc=GSE101814 | Gene Expression Omnibus, GSE101814 |

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
