## [Decision Letter]

**Decision letter after peer review:**

[Editors’ note: the authors submitted for reconsideration following the decision after peer review. What follows is the decision letter after the first round of review.]

Thank you for submitting your work entitled "Cohesin controls intestinal stem cell identity by maintaining association of Escargot with target promoters" for consideration by *eLife*. Your article has been reviewed by a Senior Editor, a Reviewing Editor, and two reviewers. The reviewers have opted to remain anonymous.

Our decision has been reached after consultation between the reviewers. While the reviewers felt that the work was potentially interesting, there was uniform agreement that although a characterisation of aneuploidy levels, mitotic errors, sister chromatid cohesion defects, etc, could be done within a few months, if the results indicate high levels of aneuploidy (which is quite likely) then the work to disentangle these two would not be trivial and would require a much longer revision time than *eLife* permits.

*Reviewer #1:*

Khaminets et al. demonstrate that Rad21, a subunit of the Cohesin complex, controls intestinal stem cell (ISC) identity by maintaining the association of Escargot (esg) with target promoters. Mosaic knockdown (KD) of Rad21 in ISCs and subsequent lineages increased the number of mature enterocytes (ECs), nuclear size and decreased the proliferative response to mild Ecc15 infection. Decreasing the loading or increasing the unloading of Rad21 in ISC lineages also resulted in similar defects. The authors show that perturbing mitotic regulators could induce ISC differentiation into ECs. However, the link between mitotic defects and possible Rad21 downregulation is not very strong. RNAseq data showed that some EC-specific genes were upregulated in Rad21 ISC KD. Furthermore, a vast majority of Esg binding to target promoter regions were absent in ISCs deficient of Rad21. Finally, the authors show that Rad21-controls ISC differentiation independent of Notch and mediates this through Esg. In summary, this is a sizable characterization the role of Rad21 in ISC differentiation despite lacking some extensive phenotypic analyses. The impact here is also limited, since most of the results of Rad21 can be explained by aneuploidy and previous studies (as mentioned by the authors) have demonstrated that aneuploidy in the *Drosophila* gut leads to ISC differentiation. As such, the reviewer is not persuaded that the present study is a strong candidate for *eLife*.

Issues to address to allow the authors to make a more convincing manuscript:

Figure 1:a) Subsection "Rad21 regulates ISC proliferation and differentiation: The authors claim that Rad21 clones had "significantly reduced size". Was this quantified? Because the Rad21 KD clones look larger (more cells/clone).

b) The increase of Pdm1/GFP-positive and nuclear size in Rad21 KD clones is not convincing in the micrographs. The internal controls (GFP-negative cells, no KD) look to have high Pdm1 percentage and larger nuclear size when compared to the GFP clones. The authors could make a more compelling argument if they compare their results with their internal controls. Furthermore, clarifying the methods might help the interpretation of these results (i.e.% of mature EC would be higher as the days after clone induction progresses).

c) Is the same defect observed is Rad21 is specifically KD in ISC (Delta[ts])?

d) Does Rad21 KD result in cell death. Does the expression of UAS-p35 inhibit this?

Figure 2:a) Similar issues with internal controls.

b) The authors did not seem to explain why CTCF was KD in the text.

Figure 3:a) What was the rationale of switching from EsgF/O (Figure 1, Figure 2, Figure 4) to ISC[ts]? For consistency perhaps Figure 1, Figure 2, Figure 4 should also show ISC[ts] results or Figure 3 should show EsgF/O data.

b) Was Pdm1/GFP-positive cells quantified for Rad21 overexpression? If so, did it show the opposite effect when compared to Rad21 KD?

c) Can UAS-Rad21-HA rescue Rad21 LOF?

Figure 4 and Figure 8:

Similar issues with Figure 1 and Figure 2. Need internal controls to make argument convincing. Can the authors elaborate why polo KD in Figure 4B has the same effect as the overexpression of the constitutively active form of Polo in Figure 2B?

Figure 5—figure supplement 1:

Is there a way to normalize your Rad21 expression to some internal control? In Figure 3C, ECs are positive for Rad21. This could be a good internal control, since the manipulation should not affect Rad21 levels in ECs.

Figure 6:

In RNAseq data, was the level of Rad21 decreased in for Rad21 KD and raised during Rad21 overexpression when compared to the control? Was there a control where there was no gene manipulation?

Materials and methods section:

For experiments that used Gal80[ts], what temperature were flies raised at? What days were they raised to the permissive temperature? For how long? What ages were the flies when the guts were dissected? What region of the midgut was analyzed for Pdm1/GFP-positive and nuclear size?

*Reviewer #2:*

In this manuscript, Jasper and co-workers report the interesting finding that cohesin is required for *Drosophila* ISCs maintenance. This role is proposed to be at least partly attributed to the role of cohesin in the maintenance of genome architecture, through the association of Esc with target promoters. Overall, this is a well performed study and although some critical controls are missing (outlined below) it provides interesting data supporting the emerging role of cohesin in gene-expression regulation and hence cell identity.

A small caveat of this study is that it remains to be addressed how much of the observed phenotypes depends on mitotic role of cohesin vs. non-mitotic roles. I agree with most of the arguments presented that sustain a non-mitotic role. However, it is odd that simple experiments to estimate the degree of mitotic errors/rate of aneuploidy/PSCS were never performed. Importantly, the actual levels of Rad21 in their RNAi experiments were never quantified. Previous studies have shown that aneuploidy induced by other means causes premature differentiation of ISCs. This work leaves open the question whether cohesin removal would promote differentiation if cells would not encounter the additional stress that mitotic errors may be imposing. Admittedly, this is not a trivial issue to access and therefore it should not compromise the publication of this work.

Essential revisions:

1) One major argument in favor of the authors hypothesis for a gene-expression role is, as the authors rightly mention, the low proliferation rate of ISCs in homeostasis conditions. However, for readers less familiar with the tissue, it would be important to put some numbers in this assessment (what is the% of cells that divide within, e.g. a 24 hour period?) and compare it with the experimental layout used (e.g. how many days after RNAi induction were the experiments performed?).

2) The authors use a quantitative argument to exclude a major contribution of mitotic defects (Figure 4), i.e., a lower degree of differentiation upon other mitotic perturbations. This is a rather weak argument. Not only the degree of RNAi depletion may vary (which was never tested) but the way such perturbations impact on mitotic fidelity is also very different. Loss of cohesion will invariably lead to far more aneuploid cells (nearly 100% in case of total loss of cohesion). The other methods applied will certainly compromise fidelity but to a much lower extent. A fair assessment of aneuploidy levels upon the used perturbations would help to clarify this issue.

3) Data in Figure 7 should include Rad21*^RNAi^* alone. With the major caveat of relying on historical controls, the numbers of Pdm1/GFP-positive cells presented in Figure 1 is higher in Rad21 alone so a partial role for Notch may not be excluded. Ideally, the authors should include this control in a paired experiment for proper comparison.

4) Similarly, in Figure 8, what is the effect of UAS-esg alone? It is not unthinkable that this may alone induce over-proliferation of ISCs even in WT conditions. If so, I am unsure what to conclude from the double RNAi experiment.

[Editors’ note: further revisions were requested prior to acceptance, as described below.]

Thank you for resubmitting your work entitled "Cohesin controls intestinal stem cell identity by maintaining association of Escargot with target promoters" for further consideration by *eLife*. Your revised article has been evaluated by Utpal Banerjee (Senior Editor) and a Reviewing Editor (Elaine Fuchs) and re-reviewed by two of our external reviewers.

The manuscript has been improved but there are just a few lingering issues from reviewer 2 that you should address textually prior to official acceptance:

*Reviewer #1:*

Revisions are fine. We recommend publication.

*Reviewer #2:*

In this revised version of the manuscript Khaminets and co-workers provide additional evidence to support that the observed premature differentiation is at least partly independent of possible mitotic defects (particularly highlighting the effect on single cell clones), and therefore suggest a gene-expression related role for cohesin in stem cell identity. Although it remains unclear the exact contribution of aneuploidy to the reported phenotype (note that non-cell autonomous mechanisms may also be in place), I think the data provided here does support that most of the observed phenotypes are likely to be attributed to gene expression changes. I am therefore in favour of its publication in *eLife*. I would just advice the authors to include some of the clarifications that are made in the rebuttal letter clearer in the original manuscript (e.g. mention clearer the single clones effect, explain the CTCF results in the context of *Drosophila* literature, etc).

---

## [Author Response]

[Editors’ note: The authors appealed the original decision. What follows is the authors’ response to the first round of review.]

We would like to thank the reviewers for the constructive critique and suggestions. We agree with the reviewers that aneuploidy induced by Cohesin depletion and its relevance in stem cell maintenance is an important aspect of our study and therefore requires additional clarification.

Two previous studies have reported conflicting results regarding the differentiation response of ISCs when aneuploidy is induced by knock down of spindle assembly checkpoint proteins: while depletion of bub3 was found to induce ISC differentiation and thus loss of ISCs (Gogendeau et al., 2015), depletion of BubR1, mad2, or mps1 all resulted in increased ISC proliferation and an accumulation of ISCs/EBs and EEs in the intestinal epithelium (Resende et al., 2018). Since depletion of all four factors results in aneuploidy, the differentiation response to Bub3 depletion seems to be a consequence of another function of Bub3, rather than the aneuploidy itself. We have now revised our Results and Discussion sections to give a more accurate account of the literature.

Based on these findings and our observations, we believe that loss of Cohesin also results in ISC differentiation independently of aneuploidy. Below, we state the main arguments for this thesis:

1) Our study shows that depleting Cohesin or promoting its release from DNA induces ISC differentiation. Multiple perturbations to that effect (two independent Rad21 shRNAs, Nipped B shRNA, T182D Polo mutant) elicited a significantly stronger differentiation phenotype compared to perturbation of other regulators of mitosis (AurB, Cdk1, Polo), indicating that a mitotic role of Cohesin could not completely account for ISC differentiation into ECs (Figure 1, Figure 2 and Figure 3). Note that in all cases we control for the efficiency of the perturbation by quantifying mitotic figures, and that in all cases there was a complete inhibition of proliferation (Figure 1D, Figure 2—figure supplement 1 and Figure 4C).

2) The fact that NippedB knockdown also results in ISC differentiation is critical, as the binding of Cohesin to chromatin after completion of mitosis is mediated in interphase by the complex containing Nipped B. These data thus suggest that Cohesin binding to chromatin in interphase is critical for ISC maintenance (Figure 2A and C). Similar important roles of Cohesin in interphase cells have been recently described in mammals (Melsenberg et al., 2019).

3) ISCs are normally quiescent in unchallenged young fly midguts and enter mitosis at a very low rate (up to approximately 10-20%) (Figure 1D, Figure 2—figure supplement 1, Figure 3B and Figure 4C). Cohesin depletion in ISCs would thus only lead to aneuploidy in the small subset of ISCs that enter mitosis. In contrast, Cohesin depletion led to differentiation of around 60-70% of all ISCs (Figure 1B), and to an almost complete elimination of Delta-positive cells (to approximately 2%) (Figure 5B). Such a drastic effect does not reflect the frequency of cells entering mitosis.

4) Our RNAseq and DamID analyses suggest that Cohesin has profound impact on the transcriptome of ISCs, and significantly affects the transcriptional program regulated by Escargot (Figure 6). Accordingly, Cohesin depletion led to loss of Esg promotor binding, and significantly reduced Esg-regulated transcripts. Esg over-expression in Cohesin-depleted cells further significantly rescued the premature ISC differentiation (Figure 8B) and restored homeostasis. It seems unlikely that these effects on Esg promoter loading and Esg-mediated gene regulation are an indirect consequence of aneuploidy. We agree that exploring a connection between aneuploidy and Esg function would be of interest generally. We do not believe, however, that such a study would be within the scope of our current report.

Based on these arguments and our data, we disagree with reviewer 1 that premature ISC differentiation after Rad21 knock down could be merely attributed to an unspecific response to aneuploidy. While we agree that Cohesin downregulation could cause aneuploidy, our data indicate that this would not fully account for the differentiation phenotype and the effects on Esg function.

We agree with both reviewers that it would not be trivial to completely separate mitotic and interphase functions of Cohesin in ISC maintenance. However, our data suggest that Cohesin regulates ISCs at least in part independently of its role in mitosis. We further believe that measuring aneuploidy in Cohesin depleted ISCs would also be non-trivial due to the high levels of polyploidization in differentiating ECs. Assessing chromosome numbers would be very difficult in this context.

Reviewer #1:Khaminets et al. demonstrate that Rad21, a subunit of the Cohesin complex, controls intestinal stem cell (ISC) identity by maintaining the association of Escargot (esg) with target promoters. Mosaic knockdown (KD) of Rad21 in ISCs and subsequent lineages increased the number of mature enterocytes (ECs), nuclear size and decreased the proliferative response to mild Ecc15 infection. Decreasing the loading or increasing the unloading of Rad21 in ISC lineages also resulted in similar defects. The authors show that perturbing mitotic regulators could induce ISC differentiation into ECs. However, the link between mitotic defects and possible Rad21 downregulation is not very strong. RNAseq data showed that some EC-specific genes were upregulated in Rad21 ISC KD. Furthermore, a vast majority of Esg binding to target promoter regions were absent in ISCs deficient of Rad21. Finally, the authors show that Rad21-controls ISC differentiation independent of Notch and mediates this through Esg. In summary, this is a sizable characterization the role of Rad21 in ISC differentiation despite lacking some extensive phenotypic analyses. The impact here is also limited, since most of the results of Rad21 can be explained by aneuploidy and previous studies (as mentioned by the authors) have demonstrated that aneuploidy in the *Drosophila* gut leads to ISC differentiation. As such, the reviewer is not persuaded that the present study is a strong candidate for eLife.

We would like to thank the reviewer for the careful and constructive critique of our manuscript. We would like to refer the reviewer to our introduction to this response letter for our argument against a simple role of aneuploidy in the differentiation observed when Rad21 is lost. Our data strongly support a role for Rad21 in interphase, and specifically in regulating Esg-mediated gene expression. We hope our edits and additions to the manuscript in this revised version help persuade the reviewer of this view.

Issues to address to allow the authors to make a more convincing manuscript:Figure 1:a) Subsection "Rad21 regulates ISC proliferation and differentiation: The authors claim that Rad21 clones had "significantly reduced size". Was this quantified? Because the Rad21 KD clones look larger (more cells/clone).

We thank the reviewer for pointing this out. The aim of the experiment was to analyze the rate of differentiation indicated by the frequency of Pdm1-positive cells in each lineage. While Cohesin-depleted clones show a significant increase in Pdm1 positive cells, it is correct that they are not in fact smaller with respect to the numbers of cells per clone (the size is about equal, with some minor variability in the exact number of cells seen in different guts (see Figure 1—figure supplement 1). The original statement was included by mistake and we have now made the appropriate corrections in the text to avoid misunderstandings (subsection “Rad21 regulates ISC proliferation and differentiation”).

b) The increase of Pdm1/GFP-positive and nuclear size in Rad21 KD clones is not convincing in the micrographs. The internal controls (GFP-negative cells, no KD) look to have high Pdm1 percentage and larger nuclear size when compared to the GFP clones. The authors could make a more compelling argument if they compare their results with their internal controls. Furthermore, clarifying the methods might help the interpretation of these results (i.e.% of mature EC would be higher as the days after clone induction progresses).

We believe there is a misunderstanding here and would like to clarify our experimental set-up and the interpretation of results. In this experiment we are using a standard lineage tracing method based on esg::Gal4 (also termed esg-FlpOut), where GFP is initially expressed in all ISCs and upon differentiation also in newly formed ECs. The GFP negative cells in each gut are all differentiated cells (ECs and EEs) and are thus not useful as ‘internal controls’ (as the reviewer points out, these have high percentage of Pdm1 expression and large nuclei, as expected). This is in contrast to the MARCM method where random clones of *rad21* RNAi expressing cells would be generated and non-marked lineages could be used as controls. We believe the reviewer refers to such an approach when stating above that we are using ‘mosaic knockdown’. In the approach we use, the controls are flies from separate crosses in which only GFP is expressed under the control of esgFlpOut. In contrast with the very low numbers of differentiated Pdm1-positive/GFP-positive cells (around 5%) in these controls, more than half (around 60%) of Rad21depleted ISCs differentiate into Pdm1-positive/GFP-positive cells (Figure 1B).

We clarified this experiment in our edited Results section and Materials and methods section as requested.

We would like to apologize for quality of Pdm1 signal. However, we could not achieve better results with any other commercially or academically available Pdm1 antibodies. Pdm1 is still one of the best markers for ECs but its usage is suffering from poor reagents. To support our conclusions, we have added additional images of such an experiment (Figure 1—figure supplement 1).

c) Is the same defect observed is Rad21 is specifically KD in ISC (Delta[ts])?

We have not used Delta[ts] in our study because this expression system is very weak and is not fully ISC specific either. We specifically chose the esg>FlpOut lineage tracing system in order to lineage trace and mark ISCderived cells regardless of their identity. Since Dl expression is lost in ISCs expressing Rad21*^RNAi^* (Figure 5), we would not be able to determine the ISC derived cells if we were using Dl-Gal4.

We have, however, used the ISC-restricted ISC^ts^ (esgGal4 combined with Su(H)Gal80) in all experiments in which ISC proliferation was. We have now added clarifications in Materials and methods section and additional figures showing that this driver recapitulates the phenotype of Rad21 knockdown seen in the esg-FlpOut experiments (Figure 1—figure supplement 1).

d) Does Rad21 KD result in cell death. Does the expression of UAS-p35 inhibit this?

Our RNAseq data do not suggest upregulation of cell death markers, nor do we see a substantial loss of ISC lineages in which Rad21 is knocked down, indicating that cell death is not relevant for the differentiation phenotype. We have clarified this in the text.

Figure 2:a) Similar issues with internal controls.

Please refer to our answer for Figure 1B above.

b) The authors did not seem to explain why CTCF was KD in the text.

We thank the reviewer for pointing this out. We have made appropriate additions in the text (subsection “Rad21 regulates ISC proliferation and differentiation”).

Figure 3:a) What was the rationale of switching from EsgF/O (Figure 1, Figure 2, Figure 4) to ISC[ts]? For consistency perhaps Figure 1, Figure 2, Figure 4 should also show ISC[ts] results or Figure 3 should show EsgF/O data.

We thank this reviewer for their questions. Most experiments were conducted using both backgrounds EsgF/O and ISC[ts]. EsgF/O was used for lineage tracing experiments to estimate the extent of cell differentiation/Pdm1 in different conditions. ISC[ts] was used for analyses of ISC proliferation/pH3 since shRNA/transgene expression is more restricted to stem cells. We have added clarifications in Materials and methods section and additional figure (Figure 1—figure supplement 1).

Rad21 over-expression strongly induces ISC proliferation but does not inhibit differentiation. We have included images of Rad21 overexpression in EsgF/O background (Figure 3—figure supplement 1).

b) Was Pdm1/GFP-positive cells quantified for Rad21 overexpression? If so, did it show the opposite effect when compared to Rad21 KD?

We have included images for Rad21 overexpression (Figure 1—figure supplement 1). Rad21 over-expression does not inhibit ISC differentiation.

c) Can UAS-Rad21-HA rescue Rad21 LOF?

We did not do this experiment as we would have to generate an RNAi insensitive over-expression construct for Rad21. However, Rad21-HA expression was verified, localized to the nucleus in ISCs, and colocalizing with chromatin (DAPI) in a way that appeared to correlate with endogenous Rad21 (Figure 3C and Figure 1—figure supplement 3 control panels). Furthermore, our RNAseq analysis confirmed over-expression of Rad21 in these flies, and showed that Rad21-HA overexpression and Rad21 KD share a considerable amount of affected genes (Figure 6).

Figure 4 and Figure 8:Similar issues with Figure 1 and Figure 2. Need internal controls to make argument convincing. Can the authors elaborate why polo KD in Figure 4B has the same effect as the overexpression of the constitutively active form of Polo in Figure 2B?

We thank this reviewer for their suggestion. Please, see the answer for Figure 1B (point to Figure 1B).

Polo KD and overexpression have similar effects in terms of ISC differentiation because both negatively affect mitosis. Balanced and timely activation of polo kinase has been reported to crucial for correct execution of mitosis (Pintard and Archambault, 2018). Polo KD has a pleiotropic effect impacting spindle formation, chromosome segregation, checkpoint activation etc., and should correlate with inhibition of AurB, Cdk1 (Pintard and Archambault, 2018). Constitutively active polo T182D over-expression has been reported to lead to Cohesin release from chromatin while also greatly delaying mitosis (up to hours) (Sumara et al., 2002; van de Weerdt et al., 2005). We used polo T182D as a tool (in addition to Nipped B KD) to investigate the role of Cohesin loading on DNA in ISC maintenance. We have clarified this in the text.

Figure 5—figure supplement 1:Is there a way to normalize your Rad21 expression to some internal control? In Figure 3C, ECs are positive for Rad21. This could be a good internal control, since the manipulation should not affect Rad21 levels in ECs.

We thank this reviewer for the suggestion. We did not use this internal control because EC progeny after Rad21 knock down could be derived from affected ISCs and thus could not serve as appropriate control.

Figure 6:In RNAseq data, was the level of Rad21 decreased in for Rad21 KD and raised during Rad21 overexpression when compared to the control? Was there a control where there was no gene manipulation?

We did indeed observe a decrease of Rad21 (vtd) transcripts in the knockdown samples (around 5 RPKM) and an increase in the overexpression samples (around 500 RPKM) compared to controls (around 50 RPKM) in RNAseq experiments (now shown in Figure 6 and Figure 6—source data 1). We also analyzed Rad21 overexpression and KD in parallel using immunostaining. We show that Rad21-HA is very well over-expressed in ISCs (Figure 3C). We are now providing images of ISC after Rad21 KD showing efficient protein downregulation (Figure 1—figure supplement 3).

Materials and methods section:For experiments that used Gal80[ts], what temperature were flies raised at? What days were they raised to the permissive temperature? For how long? What ages were the flies when the guts were dissected? What region of the midgut was analyzed for Pdm1/GFP-positive and nuclear size?

We thank the reviewer for pointing out these omissions. We have made appropriate additions in the Materials and methods section.

Reviewer #2:[…]Essential revisions:1) One major argument in favor of the authors hypothesis for a gene-expression role is, as the authors rightly mention, the low proliferation rate of ISCs in homeostasis conditions. However, for readers less familiar with the tissue, it would be important to put some numbers in this assessment (what is the% of cells that divide within, e.g. a 24hour period?) and compare it with the experimental layout used (e.g. how many days after RNAi induction were the experiments performed?).

We thank the reviewer for this suggestion. As the reviewer points out, ISCs divide at a very low rate in homeostatic conditions. The esg-FlpOut system we use actually allows determining the number of divisions that have occurred in our experiments. As shown in Figure 1A and Figure 2A, in control flies, ISCs have for the most part, not divided (single labeled cells) or divided only once (doublets) in the course of the experiments. Rad21 knockdown, Polo activation, or NippedB knockdown results in cell differentiation regardless of whether ISCs had divided (clones with more than one labeled cell, or not (single labeled cell). To highlight this point, we have now added arrows to these figures pointing to different types of clones (Figure 1A, Figure 2A). We have also included appropriate additions in the Materials and methods section and the text.

2) The authors use a quantitative argument to exclude a major contribution of mitotic defects (Figure 4), i.e., a lower degree of differentiation upon other mitotic perturbations. This is a rather weak argument. Not only the degree of RNAi depletion may vary (which was never tested) but the way such perturbations impact on mitotic fidelity is also very different. Loss of cohesion will invariably lead to far more aneuploid cells (nearly 100% in case of total loss of cohesion). The other methods applied will certainly compromise fidelity but to a much lower extent. A fair assessment of aneuploidy levels upon the used perturbations would help to clarify this issue.

We agree with the reviewer that, as originally formulated, the argument was somewhat weak. We do, however, believe that the fact that Rad21-RNAi, NippedB-RNAi, and PoloT182D clones are often single-cell clones in which the labeled cell has become polyploid and Pdm1-poitive constitutes strong evidence for differentiation that is not initiated by aneuploidy caused by an intervening mitotic division. It is further difficult to assess aneuploidy in polyploid nuclei, and we therefore have no ability to unambiguously rule out the contribution of aneuploidy. It is important to point out, however, that differentiation as a consequence of aneuploidy does not seem to be a robust phenotype: while depletion of *bub3* was found to induce ISC differentiation and thus loss of ISCs, depletion of BubR1, mad2, or mps1 all result in increased ISC proliferation and an accumulation of ISCs/EBs and EEs in the intestinal epithelium (Gogendeau et al., 2015; Resende et al., 2018). We have now discussed this in more detail in the Discussion section.

3) Data in Figure 7 should include Rad21^RNAi^ alone. With the major caveat of relying on historical controls, the numbers of Pdm1/GFP-positive cells presented in Figure 1 is higher in Rad21 alone so a partial role for Notch may not be excluded. Ideally, the authors should include this control in a paired experiment for proper comparison.

The critical point in this experiment was to show that the differentiation induced by Rad21 knockdown could still be achieved in N^RNAi^ conditions. As shown in Figure 7A (and extensively reported in the literature), N^RNAi^ results in the induction of ISC (small cell) tumors. Our data clearly show that in that context, knocking down Rad21 is sufficient to prevent ISC proliferation and cause the differentiation into Pdm1-positive cells. The effects of Rad21 alone in a wild-type context were described extensively in Figure 1 and are consistent with the phenotypes shown in Figure 7. We do not believe that adding such a knockdown in Figure 7 adds anything to the conclusion.

4) Similarly, in Figure 8, what is the effect of UAS-esg alone? It is not unthinkable that this may alone induce over-proliferation of ISCs even in WT conditions. If so, I am unsure what to conclude from the double RNAi experiment.

We thank the reviewer for this suggestion. We have included UAS-Esg alone in this experiment (Figure 8). UAS-Esg increases ISC proliferation slightly (Figure 1—figure supplement 1C). This does not change our conclusions, as Esg overexpression clearly and efficiently overcomes the premature ISC differentiation phenotype induced by Rad21 depletion, fully restoring the number of Pdm1/GFP-positive cells to the level of controls (Figure 8B).

[Editors’ note: what follows is the authors’ response to the second round of review.]

The manuscript has been improved but there are just a few lingering issues from reviewer 2 that you should address textually prior to official acceptance:Reviewer #2:In this revised version of the manuscript Khaminets and co-workers provide additional evidence to support that the observed premature differentiation is at least partly independent of possible mitotic defects (particularly highlighting the effect on single cell clones), and therefore suggest a gene-expression related role for cohesin in stem cell identity. Although it remains unclear the exact contribution of aneuploidy to the reported phenotype (note that non-cell autonomous mechanisms may also be in place), I think the data provided here does support that most of the observed phenotypes are likely to be attributed to gene expression changes. I am therefore in favour of its publication in eLife. I would just advice the authors to include some of the clarifications that are made in the rebuttal letter clearer in the original manuscript (e.g. mention clearer the single clones effect, explain the CTCF results in the context of *Drosophila* literature, etc).

We thank the reviewer for their careful reading of the manuscript, very helpful suggestions, and positive remarks in favor of publication in *eLife*.

As the reviewer suggested, we have now incorporated additional clarifications from our rebuttal letter into the manuscript:

- about the role of aneuploidy (Discussion section);

- about CTCF results (subsection “Rad21 regulates ISC proliferation and differentiation”, Discussion section);

- about ‘single cell clone effect’ (subsection “Rad21 regulates ISC proliferation and differentiation”);

-about using polo T182D (subsection “Rad21 regulates ISC proliferation and differentiation”);

about cell death markers (subsection “Transcriptional role of Rad21 in regulating ISCs”).